# Cloud droplet activation properties and scavenged fraction of black carbon in liquid-phase clouds at the high-alpine research station Jungfraujoch (3580 m a.s.l.)

Ghislain Motos[1], Julia Schmale[1], Joel C. Corbin[1a], Robin Modini[1], Nadine Karlen[1b], Michele Bertò[1], Urs Baltensperger[1] and Martin Gysel-Beer[1]

[1]Laboratory of Atmospheric Chemistry, Paul Scherrer Institute, 5232 Villigen PSI, Switzerland
[a]Now at Measurement Science and Standards, National Research Council Canada, 1200 Montreal Road, Ottawa K1A 0R6, Canada
[b]Now at Institute for Sensors and Electronics, University of Applied Sciences (FHNW), Windisch, Switzerland

*Correspondence to:* Martin Gysel-Beer (martin.gysel@psi.ch)

**Abstract.** Liquid clouds form by condensation of water vapour on aerosol particles in the atmosphere. Even black carbon (BC) particles, which are known to be little hygroscopic, have been shown to readily form cloud droplets once they have acquired water-soluble coatings by atmospheric aging processes. Accurately simulating the life cycle of BC in the atmosphere, which strongly depends on the wet removal following droplet activation, has recently been identified as a key element for accurate prediction of the climate forcing of BC.

Here, to assess BC activation in detail, we performed in-situ measurements during cloud events at the Jungfraujoch high mountain station in Switzerland in summer 2010 and 2016. Cloud droplet residual and interstitial (unactivated) particles as well as the total aerosol were selectively sampled using different inlets, followed by their physical characterization using scanning mobility particle sizers (SMPSs), multi-angle absorption photometers (MAAPs) and a single particle soot photometer (SP2). By calculating cloud droplet activated fractions with these measurements, we determined the roles of various parameters on the droplet activation of BC. The half-rise threshold diameter for droplet activation ($D_{\text{half}}^{\text{cloud}}$), i.e. the size above which aerosol particles formed cloud droplets, was inferred from the aerosol size distributions measured behind the different inlets. The effective peak supersaturation ($SS_{\text{peak}}$) of a cloud was derived from $D_{\text{half}}^{\text{cloud}}$ by comparing it to the supersaturation dependence of the threshold diameter for cloud condensation nuclei (CCN) activation measured by a CCN counter (CCNC). In this way, we showed that the mass-based scavenged fraction of BC strongly correlates with that of the entire aerosol population because $SS_{\text{peak}}$ modulates the critical size for activation of either particle type. Fifty percent of the BC-containing particles with a BC mass equivalent core diameter of 90 nm were activated in clouds with $SS_{\text{peak}} \approx 0.21$ %, increasing up to ~80 % activated fraction at $SS_{\text{peak}} \approx 0.5$ %. On a single particle basis, BC activation at a certain $SS_{\text{peak}}$ is controlled by the BC core size and internally mixed coating, which increases overall particle size and hygroscopicity. However, the resulting effect on the population averaged and on the size integrated BC scavenged fraction by mass is small for two reasons: first, acquisition of coatings only matters for small cores in clouds with low $SS_{\text{peak}}$ and, second, variations in BC core size distribution and mean coating thickness are limited in the lower free troposphere in summer.

Finally, we tested the ability of a simplified theoretical model, which combines the $\kappa$-Köhler theory with the Zdanovskii-Stokes-Robinson (ZSR) mixing rule under the assumptions of spherical core-shell particle geometry

and surface tension of pure water, to predict the droplet activation behaviour of BC-containing particles in real clouds. Predictions of BC activation constrained with $SS_{peak}$ and measured BC-containing particle size and mixing state were compared with direct cloud observations. These predictions achieved closure with the measurements for the particle size ranges accessible to our instrumentation, that is, BC core diameters and total particle diameters of approximately 50 nm and 180 nm, respectively. This clearly indicates that such simplified theoretical models provide a sufficient description of BC activation in clouds, as previously shown for activation occurring in fog at lower supersaturation, and also shown in laboratory experiments under controlled conditions. This further justifies application of such simplified theoretical approaches in regional and global simulations of BC activation in clouds, which include aerosol modules that explicitly simulate BC-containing particle size and mixing state.

## 1 Introduction

Natural and anthropogenic atmospheric aerosol particles cause a global cooling of the Earth's surface, partially compensating the warming caused by greenhouse gases (Boucher et al., 2013). Black carbon (BC), formed when fossil and biogenic fuels undergo incomplete combustion, is emitted by a large range of anthropogenic and natural sources and has unique properties leading to complex climate effects. BC is a strong light-absorber, resulting in a positive industrial-era forcing (warming) via aerosol-radiation interactions (ari; +0.71 W m$^{-2}$, 90 % uncertainty range: +0.08 to +1.27 W m$^{-2}$; Bond et al., 2013). BC can also activate to cloud droplets but can cause evaporation of droplets by releasing heat due to absorption of light; this also results in a positive industrial-era forcing via aerosol-cloud interactions (aci; +0.23 W m$^{-2}$, 90 % uncertainty range: -0.47 to +1.0 W m$^{-2}$; Bond et al., 2013). With aging, organic and inorganic matter can condense or coagulate to form a coating surrounding BC cores. This transition from external to internal mixing of BC results in two climate-relevant effects: firstly, the coating modifies the particle absorption with effects that are still under debate. Some studies reported an absorption enhancement assuming that the coating focuses the solar radiation towards the BC core. This is known as the lensing effect (e.g. Fuller et al., 1999; Bond et al., 2006). Other studies hypothesized that the coatings block the radiation, resulting in a reduction of the absorption by BC (e.g. Luo et al., 2018). Secondly, it increases the size and the hygroscopicity of the BC-containing particle, decreasing its critical supersaturation, i.e. the minimum supersaturation required for a particle to activate to a droplet. This latter effect was shown for diesel soot coated with secondary organic aerosol in a laboratory study by Tritscher et al. (2011) and also for atmospheric BC mainly coated with organic compounds (Kuwata et al., 2009). The overall climate forcing induced by the coating acquisition of BC is still poorly understood because it entangles the contributions of both aci and ari. The enhanced formation of cloud droplets increases the lifetime and the brightness of clouds (Twomey, 1974). Likely more important in most environments is that it reduces the lifetime of BC in the atmosphere by favouring its wet removal (Moteki et al., 2012), thus diminishing the time window available for absorption of solar radiation (Stier et al., 2006; Boucher et al., 2016). Lund et al. (2017) performed global model simulations testing the sensitivity of radiative forcing to the assumed threshold amount of coating needed for a BC-containing particle to be transferred from the unactivated to the activated mode. Varying this threshold resulted in changes of up to 25-50 % in ari-induced radiative forcing compared to the baseline simulation. Understanding and quantifying the links between mixing state of BC and its activation behaviour is therefore one of the main challenges that will help to assess the climate impact of BC. Alongside a better

knowledge of the preindustrial concentrations of BC, the accurate simulation of the criteria required for activation together with realistic timescales for coating acquisition will help to reduce the uncertainties related to the radiative forcing of BC.

Two main mechanisms can explain the incorporation of a particle into a droplet: impaction scavenging, which involves collision and coalescence, and nucleation scavenging, i.e. droplet activation occurring when supersaturation of the air surrounding the particle exceeds its critical supersaturation. Theoretical studies (e.g.,Flossmann and Wobrock, 2010) and field work (Ohata et al., 2016) have shown that the latter is predominant over the former, at least for accumulation mode particles, and this applies also for BC. The present study focuses on the parameters influencing the nucleation scavenging process for BC.

The Kelvin effect describes the influence of particle size on its critical supersaturation for activation as a water droplet: a large particle has a lower critical supersaturation than a smaller one with identical chemical composition. Henning et al. (2002) used *in-situ* cloud measurements at the Jungfraujoch, a high altitude site at 3580 m a.s.l. in central Switzerland, to show that the dry particle diameter is indeed the main parameter determining whether a particle activates to a droplet upon cloud formation. The threshold diameter at the Jungfraujoch is typically around 90 nm (Hoyle et al., 2016).

Raoult's law describes the influence of the chemical composition of an aqueous solution on water activity. The Köhler theory combines the Kelvin effect and Raoult's law, thereby relating particle dry size, particle composition and critical supersaturation to each other. Since many CCN closure studies confirmed the applicability of the Köhler theory to predict CCN activation of laboratory generated and ambient aerosols (e.g. Snider et al., 2003; Bougiatioti et al., 2009), Hammer et al. (2014a) proposed a method to infer the effective peak supersaturation ($SS_{peak}$) of a cloud from the droplet activation cut-off diameter observed in that cloud. In this context, $SS_{peak}$ is to be interpreted as "the maximum supersaturation encountered by a particle for a sufficiently long time that it grows into a stable droplet". We make use of $SS_{peak}$ observations to investigate the supersaturation-dependent scavenging of BC in clouds in this study.

BC is an insoluble solid; thus no reduction of water activity through Raoult's law occurs, resulting in high critical supersaturation for CCN activation at a given dry particle size (or high critical diameter for CCN activation at a given supersaturation). However, a water-soluble coating around a BC core makes it a better CCN for the size and hygroscopicity reasons explained above. Highly aged atmospheric BC has been shown to be scavenged to the same extent as the total aerosol in clouds at the Jungfraujoch (Cozic et al., 2007). However, which factors control the fraction of BC mass that activates to cloud droplets is a question that still needs to be addressed. Recently, field studies focusing on size-resolved analyses of the droplet activation behaviour of BC have tried to quantify the influence of mixing state and chemical composition on nucleation scavenging. Schroder et al. (2015) specified the minimum coating thicknesses required for droplet activation of BC in two cloud events on the Californian coast, and related it to retrieved supersaturations. However, the coating thickness calculation was heavily simplified and represented a lower limit because of technical issues, which did not allow for a comprehensive description of the conditions required for activation. Roth et al. (2016) applied single particle mass spectrometry to interstitial and cloud droplet residual particles sampled at a mountain site in central Germany (peak Schmücke; 905 m a.s.l.) to show that internally mixed inorganic salts made BC particles act as nuclei for cloud droplet formation. Zhang et al. (2017) confirmed the ability of coated BC to activate to droplets at a mountain site in Southern China (1690 m a.s.l.), and found that high fractions of sulfate in the

coatings facilitated activation compared to organic coatings, which are less hygroscopic than sulfate. While these studies provide information on parameters influencing the droplet activation of BC on a qualitative level, the relative contribution of each of these parameters to droplet activation remains to be elucidated more quantitatively. Moreover, there is a need for a direct assessment of the level of complexity that is required in the description of these parameters in order to predict droplet activation of BC and realistically simulate it in climate models.

Comparing the theoretically calculated critical supersaturation of particles that do or do not form droplets at a certain supersaturation offers the opportunity to assess the predictability of the droplet activation of BC. Such an approach was conducted in a laboratory study by Dalirian et al. (2018) who coated BC particles with known amounts of identified organic species and showed that they could accurately predict the CCN activity of the mixed particles. Matsui et al. (2013; 2016) utilized the Köhler theory considering the size and mixing state of BC-containing particles in modelling studies to show an improved simulation of BC concentrations over East Asia compared to simulations in which the mixing state was not resolved, and to observations from field measurements. However, it remains to be shown that BC activation in atmospheric clouds indeed obeys such theoretical predictions.

In this study, we selectively sampled and characterized interstitial (unactivated) particles, cloud droplet residual particles and the total aerosol (sum of interstitial plus droplet residual particles) at the high-alpine research station Jungfraujoch. Firstly, we used this approach to determine the relationship between the scavenged fraction of total BC mass and $SS_{peak}$. Secondly, we compared the observed cloud droplet activation of individual BC-containing particles with the theoretically predicted behaviour, the latter being constrained with single particle measurements of particle size and mixing state. We could show, to our knowledge for the first time for ambient cloud droplet activation of BC, that simplified $\kappa$-Köhler theory (Petters and Kreidenweis, 2007) combined with the Zdanowski-Stokes-Robinson (ZSR; Stokes and Robinson, 1966) mixing rule and assumption of a spherical core-shell morphology adequately describes the nucleation scavenging threshold of BC.

## 2 Methods

### 2.1 Measurement site

A field campaign was conducted at the high-alpine research station Jungfraujoch (3580 m a.s.l. in central Switzerland) from 12 June to 6 August 2016. Additional results are included from measurements conducted during the Cloud and Aerosol Characterization Experiment 2010 (CLACE2010) campaign at the same site during the same period of the year (19 June 2010 to 17 August 2010). The exact same instruments were used during both campaigns. Over the last 20 years, the Sphinx laboratory at the Jungfraujoch has hosted numerous field experiments on aerosol-related research (Bukowiecki et al., 2016), specifically addressing aerosol-cloud interactions during CLACE campaigns (e.g. Sjogren et al., 2008; Zieger et al., 2012), new particle formation (e.g. Bianchi et al., 2016; Tröstl et al., 2016), as well as continuous characterization of aerosol properties and trends (Collaud Coen et al., 2013). In 1995, the aerosol monitoring became part of the Global Atmosphere Watch (GAW) program of the World Meteorological Organization (WMO). Further environmental research

comprises e.g. a thorough study of the aerology and air mass dynamics around this site (e.g. Poltera et al., 2017), which is important to understand aerosol transport phenomena.

The Jungfraujoch is located on a mountain pass oriented in the direction southwest-northeast between the Jungfrau (4158 m a.s.l) and Mönch (4107 m a.s.l.) peaks. Owing to this, two main wind directions are observed from the southeast and the northwest. The relative proximity of the Jungfraujoch to lower altitude pollution sources as well as its presence within clouds about 40 % of the time (Baltensperger et al., 1997) makes it an appropriate site to study black carbon-cloud interactions. According to Herrmann et al. (2015), free tropospheric (FT) conditions prevail for 39 % of the time at the Jungfraujoch, but only around 20 % in summer. Pollution injections from the planetary boundary layer (PBL) increase the number concentration of particles larger than 90 nm from typical levels under FT conditions of around 40 cm$^{-3}$ up to 1000 cm$^{-3}$; these injections are the dominant source of CCN at the Jungfraujoch in summer, when mostly liquid clouds form (mixed-phase clouds can occur in case of low temperatures).

### 2.2 Inlets and instruments

### 2.2.1 Inlets

Aerosols were sampled through three different inlets during the whole campaign (Fig. 1): a total inlet, an interstitial inlet and a pumped counterflow virtual impactor (PCVI). We used stainless steel lines and short sections of electrically conductive tubing close to the instruments. The total inlet sampled interstitial (unactivated) particles, cloud droplets and ice crystals when mixed-phase clouds were present. This inlet was designed for sampling droplets with diameters up to 40 µm at wind speeds of up to 20 m s$^{-1}$ (Weingartner et al., 1999) and is also used for continuous GAW aerosol monitoring. It was heated to around 20 °C to decrease the relative humidity in the lines below 20 %. The interstitial inlet consisted of an aerodynamic size discriminator (Very Sharp Cut Cyclone, BGI, Butler, NJ, USA; described in Kenny et al., 2000) to sample unactivated aerosol with a flow rate of 16.7 L min$^{-1}$. Laboratory tests prior to the campaign indicated that variations of the flow rate by 1 L min$^{-1}$ had little influence on the cut-off, which varied between 2.2 and 2.4 µm. In order to characterize the different line losses between the interstitial and the total inlet, their corresponding number size distributions were compared during clear sky conditions, for which they should be identical in the submicron size range. The PCVI (Brechtel, Hayward, CA, USA) samples cloud droplets and ice crystals in case of mixed-phase clouds, and provides the dry residual particles to the aerosol instruments. Detailed information about the PCVI can be found in Boulter et al. (2006) and Kulkarni et al. (2011). All three inlets were located on the roof of the laboratory, connected to the instruments via 3 to 4 meter-long vertical lines. The redundancy of aerosol sampling by the three inlets allowed us to verify that very similar results were obtained for droplet activation behaviour by comparing concentrations and particle number size distributions of the interstitial aerosol with the total aerosol, or cloud droplet residues with the total aerosol (Sect. 3.7).

### 2.2.2 BC instruments

Measurements of the refractory BC (rBC) mass and optical sizing of BC-free and BC-containing particles were done by a single particle soot photometer (SP2 upgraded to 8-channel Revision C version, Droplet Measurement Technologies, Longmont, CO, USA). The SP2 detects incandescent and scattered light from particles passing through a high intensity intra-cavity Nd:YAG laser ($\lambda$=1064 nm). This was the only instrument switching

(automatically) between all three inlets. The SP2 provides, within its detection limits, the number size distributions and concentrations of both BC-containing and BC-free particles as well as the mass size distributions and concentrations of BC-containing particles only. Here we emphasize that "BC-free" and "BC-containing" particle are operational definitions based on whether the SP2 detects an incandescence signal or not.

Accordingly, a subset of the particles classified as "BC-free" may still contain a tiny BC core with an rBC mass equivalent diameter ($D_{rBC}$) smaller than ~50 nm. The rBC mass is inferred from the laser-induced incandescence signal empirically calibrated with fullerene soot and the optical diameter is inferred from the scattering signal calibrated with spherical polystyrene latex size standards, as described in Laborde et al. (2012b). Moreover, the thickness of the coating surrounding the BC core was retrieved under the assumption of spherical core-shell

morphology by subtracting the rBC core mass equivalent diameter from the total particle optical diameter on a single particle basis. BC-free particles do not evaporate while crossing the laser beam, making optical sizing straightforward. By contrast, BC-containing particles evaporate, which results in a perturbed scattering signal. The total particle optical diameters of BC-containing particles were thus determined by the leading edge-only (LEO)-fit method (Gao et al., 2007; Laborde et al., 2012a). Briefly, the leading edge of the scattering signal

remains unperturbed, which makes it possible to reconstruct the unperturbed maximum scattering amplitude with additional information provided by a position-sensitive split scattering detector. The reconstructed scattering amplitude is then used to infer the total particle optical diameter, for particles larger than around 180 nm only. Here we define the leading edge to be the part of the signal with an intensity less than 3 % of the maximum signal intensity. A refractive index of 1.50+0i (1.60+0i for CLACE2010 data) was chosen to convert

the scattering cross section measurements of BC-free particles to optical diameters, which brought the SP2 and SMPS derived size distributions in agreement in the overlapping size range (the optical sizing is only weakly sensitive to the choice of refractive index as shown by Taylor et al., 2015). For BC-containing particles, the same refractive index was used for the coatings, while 2.00+1.00i (2.26+1.26i for CLACE2010 data) was chosen for the BC cores. This choice resulted in agreement between the optical diameters of the bare BC cores

measured just before incandescence onset and the corresponding rBC mass equivalent diameters.

A second, qualitative method for BC mixing state analysis, classifies the BC-containing particles into two classes, one exclusively for "thickly" coated BC, the other including all remaining degrees of coating thickness from "none" through "thin" to "moderate". This "delay time" method, described in Schwarz et al. (2006), is based on the measurement of the time difference between the scattering signal peak and the incandescence

signal peak of a particle. Delay time histograms were characterized by two distinct modes corresponding to the two above-mentioned classes. The measurements of BC core mass equivalent diameter and coating thickness are based on the assumption of a spherical core and a concentric coating surrounding the core.

The incandescence and scattering detectors of the SP2 were calibrated three times during the CLACE2016 campaign: on 3 June, 17 July and 3 August 2016. A fourth calibration of the scattering detector took place on 1

July. The BC counting efficiency of the SP2 was checked against a CPC at the beginning and the end of the campaign. On 11 July, the YAG crystal had to be changed; this caused an interruption in the SP2 operation until 17 July. After that date, the SP2 was switched on only during cloud events to preserve laser power. During the CLACE2010 campaign, the scattering detector was calibrated four times: on 16 and 27 January, 8 February and 3 March. The incandescence detector was calibrated on 27 January.

Multi-angle absorption photometers (MAAP model 5012, Thermo Fisher Scientific, Waltham, MA, USA) were installed downstream of the total and interstitial inlets (two MAAPs in total). This instrument determines the aerosol absorption coefficient at a wavelength of 637 nm by collecting particles on a fiber filter and measuring the transmission and back scattering of laser light at multiple angles (Petzold and Schönlinner, 2004). The firmware output at 1-minute time resolution of the equivalent black carbon (eBC) mass concentration was used, which is calculated from the measured absorption coefficient using a mass absorption cross-section (MAC) value of 6.6 $m^2$ $g^{-1}$. The agreement between the two MAAPs was checked during out-of-cloud conditions and discrepancies were less than 5 % throughout the whole campaign.

### 2.2.3 Particle number concentration and size distribution

One condensation particle counter (CPC, TSI Inc., Shoreview, MN, USA) was installed downstream of each inlet in order to measure the particle number concentration. Three different CPC models were used: 3010 for the interstitial inlet (with a 50 % detection efficiency reached at 10 nm), 3022 for the PCVI (7 nm) and 3025 for the total inlet (3 nm). The quality of total and interstitial CPC data was controlled by comparing them during out-of-cloud conditions.

Two custom-built SMPS systems, each consisting of a differential mobility analyzer (DMA, model TSI long, TSI Inc., Shoreview, MN, USA) and a CPC (TSI model 3775 for the total inlet and 3022A for the interstitial inlet), measured aerosol number size distributions at a time resolution of 6 minutes. The range of measured mobility diameter extended from 22 to 604 nm. One SMPS was placed downstream of the total inlet (it is used for the continuous GAW measurements) while the other switched every 12 min (2 scans) between the interstitial inlet and the PCVI. The sizing and counting efficiencies of both SMPS systems were checked using 150 nm and 269 nm polystyrene latex spheres (PSL) every two weeks during the campaign. Quality assurance further included an intercomparison of all 5 CPCs at the beginning and at the end of the campaign: their number concentration readings agreed within 10 %.

### 2.2.4 CCNC

Cloud condensation nuclei number concentrations in polydisperse aerosol samples were measured at four different supersaturations (0.35; 0.40; 0.50 and 0.70 %; total measurement cycle of 225 min) with a cloud condensation nuclei counter (DMT model CCN-100, Droplet Measurement Technologies, Longmont, CO, USA; see details in Roberts and Nenes, 2005). Calibrations of the CCNC took place on 10 June and 4 August 2016 and gave very similar results, with less than 5 % difference between the supersaturation calibration curves. Concerning the CLACE2010 campaign, the CCNC was calibrated on 16 June 2010.

### 2.2.5 Cloud microphysics and meteorological parameters

In order to detect the presence of clouds and measure the liquid water content (LWC), a particulate volume monitor (PVM-100, Gerber Scientific Inc., Reston, VA, USA; described in Gerber, 1991) was installed on the roof of the laboratory, at the same height and around 3 m away from the inlets. The PVM detects light scattered by the cloud droplets in forward direction at multiple angles to infer the LWC. It was calibrated every week with a calibration disk provided by the manufacturer.

Measurements of air temperature 2 m above the ground, wind speed and direction are continuously conducted at the Jungfraujoch and are part of the SwissMetNet network of MeteoSwiss.

## 3 Theory and data analysis approaches

### 3.1 Identification of cloud events and stable cloud periods

The occurrence of in-cloud conditions during the campaign was determined with the LWC measurements of the PVM. The criterion for defining a cloud event was a minimum LWC of 0.1 g m$^{-3}$ for at least one hour. Cloud events typically lasted 2 to 15 hours. Very short periods (a few minutes) during the cloud events, during which the LWC dropped below 0.1 g m$^{-3}$, were likely caused by entrainment of dry air near the edge of a cloud and were excluded from the analysis. Cloud presence was independently confirmed based on significant differences

in the particle number concentrations measured behind the total and interstitial inlets. In total, 24 cloud events were sampled in 2016 (see Table S1).

As discussed below, aerosol hygroscopicity and cloud peak supersaturation often varied substantially over the full duration of a cloud event. Therefore, stable periods within a cloud event were identified as periods with limited variability in key aerosol and cloud parameters. Sometimes even two distinct stable periods were

identified in a single cloud event, resulting in a total of 11 "stable cloud periods" from the CLACE2016 campaign which were chosen for further detailed analysis (see Table 1). The analyses of three stable cloud periods extracted from the CLACE2010 campaign are also shown. Combining both campaigns, these periods add up to a total duration of 14.1 hours.

### 3.2 Activated fraction, activation diameters and activation plateau

For in-cloud conditions, we define the size-dependent activated fraction, $AF(D_X)$, as the number fraction of particles with diameter $D_X$ that formed a cloud droplet. The activation spectrum is inferred from size distribution measurements behind two different inlets. A first option is to use the interstitial (int) and total inlets (tot), as e.g. done by Hammer et al. (2014b):

$$AF_{int}(D_X) = \frac{\frac{dN_{tot}}{dlogD_X}(D_X) - \frac{dN_{int}}{dlogD_X}(D_X)}{\frac{dN_{tot}}{dlogD_X}(D_X)} \quad (1)$$

Here, $\frac{dN_Y}{dlogD_X}$ is the particle number size distribution with respect to diameter $D_X$ measured behind the inlet type "Y", where "X" is a placeholder indicating the diameter type, i.e. dry particle mobility diameter, dry particle optical diameter or BC core mass equivalent diameter.

Alternatively, the activation spectrum is inferred from the data measured behind the PCVI inlet and the total inlet:

$$AF_{PCVI}(D_X) = \frac{\frac{dN_{PCVI}}{dlogD_X}(D_X)}{\frac{dN_{tot}}{dlogD_X}(D_X)} \quad (2)$$

The activated fraction typically follows an S-shape curve, as seen in Figures 2 and 3, starting with null activation at sufficiently small diameter and reaching a plateau value at sufficiently large diameters. We define

the threshold dry diameter for activation to cloud droplets, $D_{\text{half}}^{\text{cloud}}$, as the diameter at which the activated fraction reaches half of the activated fraction at the plateau.

We use the term "activated fraction" in the context of particle number, whereas we use the term "scavenged fraction" when presenting mass fractions of particulate matter incorporated into cloud droplets relative to the total aerosol. Size-resolved activated fraction and scavenged fraction of BC are identical in the special case of choosing the BC core mass equivalent diameter for the diameter scale. However, activated fraction and scavenged fraction integrated over a certain diameter range are not identical due to the different size dependence of the weighting factor when integrating number or mass.

### 3.3 $\kappa$-Köhler theory

The equilibrium size of an aerosol particle under subsaturated relative humidity (RH) conditions, and its activation threshold to a cloud droplet at supersaturated RH conditions, depend primarily on the particle dry diameter ($D_{\text{dry}}$) and chemical composition. The Köhler theory (Köhler, 1936) describes equilibrium vapour pressure (RH$_{\text{eq}}$) over a solution droplet by combining the Kelvin effect, capturing the influence of size, and Raoult's law, capturing the influence of particle composition. Parameterizing Raoult's law is complicated by non-ideal interactions between water and solutes and by the presence of both inorganic and organic compounds in internally mixed particles. Petters and Kreidenweis (2007) proposed the use of a single hygroscopicity parameter $\kappa$ to describe Raoult' law, i.e. to describe the dependence of water activity on solution concentration for a given particle composition. This approach, commonly referred to as the $\kappa$-Köhler theory, allows the equilibrium supersaturation over the solution to be expressed as a function of droplet diameter, $D_{\text{drop}}$:

$$SS_{\text{eq}}(D_{\text{drop}}) := RH_{\text{eq}}(D_{\text{drop}}) - 1 = \frac{D_{\text{drop}}^3 - D_{\text{dry}}^3}{D_{\text{drop}}^3 - D_{\text{dry}}^3(1-\kappa)} \exp\left(\frac{4 \cdot \sigma_{s/a} M_{\text{w}}}{RT \rho_{\text{w}} D_{\text{drop}}}\right) - 1 \qquad (3)$$

Where $\rho_{\text{w}}$ and $M_{\text{w}}$ are the density and the molar mass of water, respectively, $\sigma_{s/a}$ is the surface tension of the solution-air interface (hereafter assumed to be equal to the surface tension of pure water), $R$ is the universal gas constant and $T$ is the absolute temperature. From Eq. 3, it follows that $SS_{\text{eq}}$ has a global maximum at the droplet diameter above which the particle is considered to be activated to a cloud droplet. Equation 3 further implies an unambiguous relationship between the dry diameter of a particle, its hygroscopicity parameter $\kappa$ and the maximum supersaturation. Knowing two of these parameters makes it possible to infer the third one using Eq. 3 and numerical approaches. For example, knowing the supersaturation and $\kappa$ implies a critical dry diameter above which all particles activate to cloud droplets.

### 3.4 Retrieval of the aerosol hygroscopicity parameter $\kappa$

The combination of total CCN number concentration at a defined supersaturation, measured by the CCNC in polydisperse set-up, with total particle number size distribution, measured by the SMPS, makes it possible to infer the critical dry diameter of the ambient aerosol for the supersaturation set in the CCNC (Kammermann et al., 2010b). This approach was applied for the first time at the Jungfraujoch by Jurányi et al. (2011) under the assumption that the aerosol is internally mixed. Specifically, the particle number size distribution was integrated from the maximum diameter to the diameter at which the integrated particle number concentration is equal to the simultaneously measured CCN number concentration. The lower limit of integration matching this condition

corresponds to the critical dry diameter for CCN activation, $D_{\text{crit}}^{\text{CCN}}$. As the CCNC was repeatedly stepped through a sequence of 4 different supersaturations between 0.70 % and 0.35 %, this provided the relationship between supersaturation and corresponding critical dry diameter for CCN activation of the ambient aerosol. This was converted to the corresponding relationship between particle dry diameter and the hygroscopicity parameter $\kappa$ based on Eq. 3. Figures 4a and 4b show these relationships for two example cloud periods. The data points are located along the dotted lines because only two free parameters are left in the $\kappa$-Köhler theory for a fixed supersaturation applied in the CCNC.

**3.5 Retrieval of cloud effective peak supersaturation**

The activation of aerosol particles in an ambient cloud depends on the peak supersaturation reached during cloud formation. We applied the method introduced by Hammer et al. (2014a) to retrieve the effective peak supersaturation ($SS_{\text{peak}}$) of clouds observed at the Jungfraujoch. Briefly, the half-rise threshold diameter for activation to cloud droplets ($D_{\text{half}}^{\text{cloud}}$) during a cloud event was inferred from the SMPS particle number size distribution measurements behind the total and interstitial inlets as explained in Sect. 3.2 and shown in Figure 2. The corresponding $SS_{\text{peak}}$ of a cloud was then retrieved by inputting this $D_{\text{half}}^{\text{cloud}}$ and the corresponding CCNC-retrieved aerosol hygroscopicity parameter $\kappa$ (Sect. 3.4) into the $\kappa$-Köhler equation (Eq. 3). In the present study, the hygroscopicity parameter was, on average, found to be independent of particle size (Fig. 6), while it varied in time (Fig. 5b). Therefore, we simply considered the moving average, $\kappa_{\text{smooth}}$ (black open circles in Figure 5b), of the time-resolved $\kappa$ values at different particle sizes to be representative of the $\kappa$ value at the diameter $D_{\text{half}}^{\text{cloud}}$. This is slightly different from the approach chosen by Hammer et al. (2014a) and Motos et al. (2019), in which case the size dependence was considered by interpolation or extrapolation of size-dependent $\kappa$ values to the diameter $D_{\text{half}}^{\text{cloud}}$ (at the expense of time-averaging over a whole CCNC measurement cycle).

The apparently circular calculation in above approach, i.e. going to hygroscopicity parameter and backing it out again, is required to account for the temperature difference between droplet formation processes occurring in the CCN counter and at cloud base. Cloud base temperature, $T_{\text{cloudbase}}$, which was used as input to the $\kappa$-Köhler equation during CLOUD2016 but not CLOUD2010 because of the need of a PVM, was not directly measured. Instead it is chosen to be equal to the dewpoint corresponding to the total water content at the Jungfraujoch with a correction for the pressure difference between the Jungfraujoch and the cloud base (see Hammer et al., 2014a).

**3.6 Calculation of the critical supersaturation of individual BC-containing particles**

The critical supersaturation for droplet activation is calculated for individual BC-containing particles detected by the SP2 behind the different inlets, following the approach described in Motos et al. (2019). Briefly, the approach entails combining $\kappa$-Köhler theory (Sect. 3.3) with the ZSR mixing rule as introduced by Petters and Kreidenweis (2007). The calculation requires the total particle diameter, the volume fractions and corresponding hygroscopicity parameters of each of the components of the particle. We treat the BC-containing particles as two-component particles with total size (and volume) taken from the optical diameter of the particle measured with the SP2 (Sect. 2.2.2). The BC volume is inferred from the rBC mass measured by the SP2. The $\kappa$ value of BC is chosen to be zero as BC is insoluble. The $\kappa$ value of the coating is assumed to be equal to the mean $\kappa$ value inferred from the CCNC measurements at all supersaturations during the cloud event under investigation (the size dependence of the $\kappa$ values inferred by the process described in Sect. 3.4 was typically small, thus

justifying this type of averaging). The $\kappa$ value of the mixed particle is calculated as the volume-weighted mean of the $\kappa$ values of the BC core and its coating (ZSR mixing rule). For the CLACE2010 campaign, the $\kappa$ values were taken from the annual data set of $\kappa$ values at the Jungfraujoch published by Jurányi et al. (2011), with interquartile-ranges used for error propagation. These theoretical calculations of BC-containing particle critical supersaturation are based on the assumption of spherical core-shell morphology – a simplification that is tested in the BC activation closure presented below.

### 3.7 Correction of data downstream of the PCVI and comparison with interstitial inlet data

The PCVI, described in Sect. 2.2.1, was not operational during all but two cloud events due to technical issues linked to flow rate adjustments and icing of the inlet (Table S1). During the cloud events on 22 July and 4 August, when the PCVI functioned, the input and output flow rates were set to 11.8 L min$^{-1}$ and 1.5 L min$^{-1}$, respectively, resulting in a calculated particle enrichment factor in the outflow of 7.9 (ratio of input to output flow rate). Particle losses along the PCVI inlet line were estimated to be as high as 67 % based on total concentration measured by the CPC behind the total inlet compared to the concentration measured with the PCVI-inlet CPC when bypassing the PCVI during out-of-cloud conditions. The transmission efficiency of the PCVI was around 45±5 %, based on laboratory tests prior to the campaign. These PCVI corrections for enrichment factor, line losses and transmission efficiency were applied to the raw number size distributions measured behind the PCVI, as illustrated in Figure 3a. The absolute uncertainties in the corrected cloud droplet residual number size distributions were considerable as the loss corrections amount to a factor of ~7 in total. Nevertheless, good agreement with the total number size distribution was reached for large particle diameters, where most particles are activated to cloud droplets. Furthermore, the size-dependent activated fractions inferred from PCVI and total inlet data agreed well with those inferred from the interstitial and total inlet data (Eqs. 1 and 2, respectively; shown in Figures 3b and S2b). This shows, firstly, that the PCVI and the interstitial inlets indeed sampled exclusively cloud droplet residual and interstitial particles, respectively, and secondly, that the interstitial- and PCVI-based approaches provided consistent results with respect to measured threshold diameters for activation to cloud droplets.

### 3.8 Measurement uncertainties

In the present work, the uncertainties associated to MAAP- and SMPS-derived scavenged fractions are based on propagating differences between measurements conducted behind the interstitial and total inlets during out-of-cloud conditions, immediately before and after each cloud event. The same approach applies for SMPS-derived activated fractions of the total aerosol (and the corresponding activation diameters $D_{\text{half}}^{\text{cloud}}$ and $D_{50}^{\text{cloud}}$) and SP2-derived activated fractions of BC-free particles. Uncertainties related to the SP2-derived activated fractions of BC-containing particles result from the propagation of Poisson-based counting uncertainties related to the BC core number size distributions with a sample volume uncertainty assumed to be 5 %.

The uncertainties in the retrieval of $\kappa_{\text{smooth}}$ were estimated using the deviation of $\kappa$ values around the moving average of these values representing $\kappa_{\text{smooth}}$ (see Sect. 3.5 and Figure 5). A Monte Carlo simulation was used to propagate the uncertainties estimated for $D_{\text{half}}^{\text{cloud}}$ and $\kappa$ to $SS_{\text{peak}}$. These uncertainties are listed in Table 1.

## 4 Results and discussion

### 4.1 Overview of cloud and aerosol properties

The aerosol properties observed during this study will not be discussed in detail as several comprehensive data sets of the Jungfraujoch aerosol observations have already been published (e.g. Bukowiecki et al., 2016 and references therein). By contrast, only limited data on BC properties have been published at the Jungfraujoch so far (e.g. Liu et al., 2010; Kupiszewski et al., 2016), but a more comprehensive manuscript on this topic is currently in preparation (Motos et al., in prep.). Here, we focus only on the aerosol properties that are directly relevant for determining the activation behaviour of BC in clouds.

The hygroscopicity parameter $\kappa$ was inferred for the bulk aerosol from the polydisperse CCN measurements combined with the size distribution measurements using the method described in Sect. 3.4. This provides a time series of $\kappa$ values as shown in Figure 5b for the 31 July-1 August cloud event. These $\kappa$ values are representative of different particle diameters because the CCN measurements were done at different supersaturations. The aerosol hygroscopicity varied during the cloud event. However, the short-term fluctuations were sometimes dominated by random noise due to limited counting statistics from the low particle number concentrations, e.g. during the first part of the cloud event shown in Figure 5 from about 16:00 to 22:00. By contrast, the size dependence at a given time was often small, e.g. during the last part of the 31 July-1 August event after 02:00 on 1 August. Results for two more cloud events are shown in Figure 4. The first event (Fig. 4a) is an example where $\kappa$ varied strongly over the cloud period, while the second event (Fig. 4b) is representative of the conditions during the campaign, i.e. rather stable hygroscopicity. In both of these and other cases, temporal variability typically dominated over size dependence. This justifies the use of a simple running mean of $\kappa$ data points from all supersaturations (i.e. neglecting the size dependence) in the further data analysis of droplet activation processes in clouds (see also Sect. 3.5).

The statistics of aerosol hygroscopicity over the whole CLACE 2016 campaign are shown in Figure 6. Mean values of $\kappa \approx 0.25$ and the inter-quartile range overlap well with the range of aerosol hygroscopicity observed at the Jungfraujoch during previous long-term campaigns (Kammermann et al., 2010a: mean $\kappa$ of 0.24 over 13 months; Jurányi et al., 2011: mean $\kappa$ of 0.20 over 17 months; Schmale et al., 2018: mean $\kappa$ of 0.29 over 35 months). This indicates that CLACE 2016 was representative in terms of aerosol hygroscopicity.

The size distribution measurements behind the total and interstitial inlets were used to determine the size-resolved fraction of particles that activated to cloud droplets during cloud events (Eq. 1; Sect. 3.2). The half-rise threshold diameter ($D_{\mathrm{half}}^{\mathrm{cloud}}$) for droplet activation was then inferred from these data as illustrated in Figure 2. The resulting time series of $D_{\mathrm{half}}^{\mathrm{cloud}}$ during the 31 July-1August cloud event is shown in Figure 5c, with 6 minutes time resolution. Large variations of $D_{\mathrm{half}}^{\mathrm{cloud}}$, by a factor of 2 or more, were found during this and most other cloud events.

The $\kappa$-Köhler theory was then used to infer the effective cloud peak supersaturation ($SS_{\mathrm{peak}}$) from the time-resolved $D_{\mathrm{half}}^{\mathrm{cloud}}$ and $\kappa$ values (see Sect. 3.6 for details). The variations in $D_{\mathrm{half}}^{\mathrm{cloud}}$ translate to variations in $SS_{\mathrm{peak}}$ by factors of up to 4 during individual cloud events (see Figure 5d for the 31 July-1 August cloud event). Such variations in $SS_{\mathrm{peak}}$ are primarily driven by variations in atmospheric dynamics (i.e. updraft) at the cloud base

and to a lesser extent by the number concentration of potential CCN, as demonstrated for the Jungfraujoch site by Hoyle et al. (2016). Variations in effective cloud peak supersaturation are *a priori* unrelated to the cloud LWC (Fig. 5a and Table 1), which depends only on the air parcel's height above the cloud base. The tendency of convective clouds (mostly of the *cumulus* type) to create highly spatially and temporally variable

supersaturations was reported by Politovich and Cooper (1988). Such clouds form by convection of warm air in contact with mountain faces and are often found in mountainous regions in summer.

Table S1 shows that the variations in $SS_{peak}$ were large during all cloud events. In order to study the activation of aerosol particles to cloud droplets under well-defined conditions, only continuous in-cloud periods with limited variability in $SS_{peak}$ were retained for further analysis. Additional criteria were sufficiently stable values of

hygroscopicity, particle number concentrations and BC mass concentration. A total of 14 "stable" cloud periods adding up to a total of 14.1 h of in-cloud measurements were identified (11 from CLACE2016, 3 from CLACE2010).

The size-resolved activated fractions averaged separately for all stable cloud periods of the CLACE2016 campaign are plotted in Figure 7. The mean peak supersaturation, indicated for each period by the line colour,

decreases monotonically with increasing activation threshold diameter. This is not surprising as the $SS_{peak}$ value of each stable cloud period was calculated from the $D_{half}^{cloud}$ value. However, it does imply that variations in $D_{half}^{cloud}$ were mainly driven by variations in updraft velocities and resulting supersaturations, whereas differences in aerosol hygroscopicity only caused minor additional modulation of $D_{half}^{cloud}$. The key parameters for each selected cloud period are summarized in Table 1, including the droplet number concentration inferred from the

difference in particle number concentration between the total and the interstitial inlet, and the median number concentration of potential CCN, i.e. particles with a mobility diameter larger than 90 nm ($N_{90}$; e.g. Hammer et al.; 2014a). $SS_{peak}$ ranged from 0.15 % to 0.96 % and $D_{half}^{cloud}$ from 39 nm to 150 nm. The selected periods of in-cloud measurements are representative of clouds typically encountered at the Jungfraujoch in summer. This is shown with Figure S1, in which the $SS_{peak}$ values observed during CLACE2016 are compared with results from

previous studies at the Jungfraujoch. The distribution of $SS_{peak}$ during CLACE2016 largely overlaps with the temporally more extensive observations reported by Hammer et al. (2014a). Furthermore, the systematic difference between northwestern and southeastern wind direction, explained with differences in orographic uplifting by Hammer et al. (2014a), was also found during CLACE2016.

## 4.2 Bulk analysis of BC scavenging

The scavenging of BC, i.e. the mass fraction of BC incorporated into cloud droplets, has previously been investigated at the Jungfraujoch. Cozic et al. (2007) applied the same combination of interstitial and total inlets to determine the scavenged fraction of BC (based on eBC mass measured by two MAAPs), as well as the scavenged fraction of the total aerosol (derived from SMPS measurements). They found close agreement between the scavenged fractions of BC and that of the total aerosol for warm clouds with temperature at

Jungfraujoch ($T_{JFJ}$) above -5 °C, i.e. high correlation and almost identical values on average. Such close agreement is *a priori* not expected because BC is insoluble in water; however, the authors attributed it to the high degree of internal mixing of BC in the aged aerosol at the Jungfraujoch. Figure 8a presents an equivalent analysis of BC and total aerosol volume scavenged fractions for the CLACE2016 data set of this study, which confirms the close agreement. Note that the MAAP-derived eBC mass-based scavenged fractions are consistent

with SP2-derived rBC mass scavenged fractions. However, the eBC data were chosen for Figure 8 as they are available at higher time resolution because the SP2 was switched between 3 inlets.

Going a step further, we examined the dependence of the scavenged fractions on $SS_{peak}$ by colouring the data points in Figure 8a accordingly. The fact that data points are systematically ordered by colour indicates that the scavenged fractions of both BC and the total aerosol volume are primarily controlled by variations in $SS_{peak}$, as previously suggested by Ohata et al. (2016). Indeed, plotting the same data set as scavenged fraction against $\log(SS_{peak})$ in Figure 8b reveals an "S"-shaped relationship for both the total aerosol volume and BC, which is well represented by manually fitted Hill equations (green and black dashed lines). Scavenged fractions are on average equal for BC and the total aerosol, as already shown in Figure 8a, and reach values of 50 %, 75 % and >90 % at supersaturations of 0.13-0.17 %, 0.25-0.31 %, and >0.55 %, respectively. This implies that the peak supersaturation at cloud formation must be considered to correctly describe the fraction of BC incorporated into cloud droplets through nucleation scavenging, as also shown by Ching et al. (2018) using particle-resolved model simulations. For example, the systematic difference in the mean $SS_{peak}$ between northwesterly and southeasterly wind conditions at the Jungfraujoch, as shown in Figure S1, results in systematic differences for the scavenged fractions of both BC and the total aerosol volume. This result also confirms that nucleation scavenging is the dominant mechanism resulting in the incorporation of particles (BC-free or BC-containing) into cloud droplets at the Jungfraujoch. If impaction, a process unrelated to $SS_{peak}$, were to dominate, we would not observe such a relationship between the scavenged fractions and $SS_{peak}$.

The SS-dependence of the scavenged fraction of a hypothetical, internally mixed aerosol with lognormal size distribution and size-independent hygroscopicity (composition) will follow a Hill-curve such as the dashed lines in Figure 8b. Variations in modal size would shift the position of the Hill curve, whereas deviations from lognormal size distribution shape would distort the shape of the curve. Similarly, variations and size dependence of aerosol hygroscopicity would also modulate the scavenging curve, but are probably too small to cause modifications to the same extent. Such variations are the reasons for the substantial scatter around the Hill curves in Figure 8b. The reverse conclusion is that size distribution and mean hygroscopicity must be known to accurately describe the supersaturation dependence of the scavenged fraction.

The scavenged fraction of BC mass is only expected to be equal to the total aerosol volume scavenged fraction for all peak supersaturations if BC contributes an equal fraction to the aerosol volume at any particle size and if the critical activation diameters of the BC-containing particles and total aerosol are equal. While the latter condition is closely fulfilled if BC is internally mixed with substantial coatings, size-independent BC volume fractions are *a priori* not expected. Nevertheless, the scavenged fractions of total aerosol volume and BC mass are essentially equal on average. However, deviations of several data points in Figure 8a from the "1:1"-line are greater than the measurement uncertainty, indicating that even at remote locations the BC scavenged fraction can differ from the total aerosol volume scavenged fraction in individual cloud events. This is likely due to some size dependence of the contribution of BC to the aerosol volume and/or disagreement between the critical activation diameters of BC-free and BC-containing particles. For example, new particle formation events followed by growth to sizes remaining below the droplet activation cut-off diameter, as e.g. reported in Tröstl et al. (2016), is one possible mechanism that can result in the BC scavenged fraction becoming greater than that of the total aerosol volume.

The scavenged fractions of BC and the aerosol volume observed in fog events in Zurich by Motos et al. (2019) are also included in Figure 8b. The peak supersaturation in fog is much lower than that of typical clouds at the Jungfraujoch. Nevertheless, one cloud during the CLACE 2016 campaign had a peak supersaturation as low as the fog data (~0.05 %). The scavenged fractions of BC and total aerosol volume are very low at such low peak

supersaturations because the activation cut-off is in the upper tail of the size distributions. However, during the fog events, the scavenged fraction of BC was found to be consistently lower by a factor of ~2 compared to that of the total aerosol volume. The presence of externally mixed fresh BC-containing particles at the urban location might explain this difference. Below we will show that the BC mixing state is indeed among the parameters controlling activation on the single particle level. However, more comprehensive data sets would be required to

confirm that differences in resulting bulk scavenged fractions, which also depend on size distribution and "mean" mixing state, are indeed systematic in urban fog.

Matsui (2016) used a mixing-state-resolved 3D model to simulate the mixing state of BC-containing particles over East Asia and to estimate the critical supersaturation required for CCN activation of these particles. He concluded that almost all BC-containing particles activate to form droplets at 1.0 % supersaturation while 20 to

50 % by number stay in the interstitial phase at 0.1 % supersaturation. He applied a theoretical approach equivalent to the one verified in the present study (see Sect. 4.4). These model results are in qualitative agreement with our observations. However, a direct, quantitative comparison of number-based activated fractions of BC over East Asia with mass-based scavenged fractions of BC at the Jungfraujoch is not justified because mass-and number-based activated fractions can differ for the same aerosol population, and because the

sources and levels of air pollution are different in Central Europe and East Asia.

**4.3 Influence of core size and mixing state on the activation of BC in clouds**

The SMPS measurements behind the interstitial and total inlets and Eq. 1 were used to infer the size-dependent activation of aerosol particles as discussed above in Sect. 4.1 and shown in Figure 2. The SP2 measurements behind these inlets were used in an equivalent manner to specifically investigate the activation of BC-containing

particles to cloud droplets. Figure 2 shows, on the basis of four example cloud events, that the SP2-based results for BC-free particles (blue and green lines) agree with the SMPS-based results for all particles (pink lines). This comparison is appropriate as the BC-free particles represent around 70-95 % of all particles by number (see Sect. 2.2.2 for our operational definition of "BC-free particle"). These independent measurements of activated fractions agree well because the optical diameters provided by the SP2 for BC-free particles are equal to the

respective mobility diameters measured by the SMPS, which was tested by comparing corresponding size distributions from these two instruments (not shown). Such consistency in sizing is expected for spherical particles if appropriate calibration and data processing procedures are applied to the SP2 light scattering signals. The optical sizing of BC-containing particles by the SP2 requires the more sophisticated LEO-fit technique (see Sect. 2.2.2), which was limited to optical diameters greater than 180 nm. The SP2/LEO-fit derived size-

dependent activated fractions of BC-free and BC-containing particles shown in Figure 2 as green and black lines, respectively, are in agreement within experimental uncertainty. Such agreement indicates that the majority of the BC-containing particles with a diameter greater than 180 nm consist of small BC cores with substantial coating acquired through various processes during atmospheric transport to the remote Jungfraujoch site (through e.g. condensation of volatile organic compounds, coagulation with particles or in-cloud processes).

Such small insoluble cores hardly alter the hygroscopicity of the entire particle compared to a BC-free particle. Using single particle mass spectrometry, Zhang et al. (2017) performed an equivalent comparison in Southern China and also found that the activated fraction of BC-containing particles was similar or slightly lower compared to that of the total aerosol in the vacuum aerodynamic diameter range from about 200 to 1300 nm.

The scavenged fraction of BC mass can be more directly understood by analyzing activated fractions as a function of BC core size rather than total particle diameter. The finding that the BC scavenged fraction is primarily controlled by cloud peak supersaturation, as shown in Figure 8 and discussed in Sect. 4.2, is also clearly shown in Figure 9a, which shows the BC activated fraction as a function of the rBC mass equivalent

diameter for all stable liquid cloud periods: the activated fraction increases with increasing $SS_{peak}$ at a given BC core diameter. The 2b August stable cloud period seems to be an outlier as the activated fraction is particularly high in relation to the corresponding $SS_{peak}$. However, the fact that the activation plateau was particularly high during this period explains this singularity (see Figure **Error! Reference source not found.**b). 50 % activation is reached at a BC core diameter of approximately 100 nm during the cloud periods with lowest supersaturations

of around 0.2-0.25 %. By contrast, the activated fraction is ≥90 % all the way down to $D_{rBC} \approx 60$ nm for the clouds with the highest peak supersaturations (> 0.7 %). This explains why the mass fractions of scavenged BC at the Jungfraujoch vary between roughly 60 % to 100 % for $SS_{peak}$ ranging from 0.2 % to 1 % (Fig. 8) given the fact that the BC core mass size distribution typically peaks in the range 120 nm ≤ $D_{rBC}$ ≤ 170 nm (illustrated by the mode shown in Figure 9b and Motos et al., in prep.).

Mixed-phase or even completely glaciated clouds may occur at lower temperatures. Mixed-phase clouds may result in the conversion of particles from droplets (activated particles) to interstitial aerosol through the Wegener-Bergeron-Findeisen process (e.g. Cozic et al., 2007), thereby potentially obscuring the causal relationship between $SS_{peak}$ and droplet activation. However, Verheggen et al. (2007) showed that $D_{half}^{cloud}$ remains well-defined and that only small differences in average $D_{half}^{cloud}$ exist between mixed-phase and liquid

clouds. This suggests that the Wegener-Bergeron-Findeisen process does not affect the inferred $SS_{peak}$. The fact that the Wegener-Bergeron-Findeisen evaporates some cloud droplets, whereby the droplet nuclei are released back to the interstitial aerosol, explains that the activated fraction was lower in most clouds at temperatures below -5 °C compared to that in warm clouds at comparable $SS_{peak}$ (see the dashed lines in Figure **Error! Reference source not found.**a). The 26-27b June stable cloud period is an exception in so far as the activated

fraction of BC was comparable to warm cloud results despite low $T_{JFJ}$. This cloud may have been a supercooled liquid cloud rather than a mixed-phase cloud. The fact that a supercooled liquid cloud was an exception at the Jungfraujoch for temperatures below -5 °C may indicate that cloud glaciation is only rarely limited by ice nuclei concentrations. No *in-situ* measurements of cloud properties were performed to determine the phase of the clouds in order to test this hypothesis.

According to the Köhler theory (Sect. 3.3), the BC core diameter of an internally mixed BC-containing particle is not the decisive parameter for its critical supersaturation (even for a hypothetical spherical core-shell morphology). Instead, in the absence of surfactants, the overall particle diameter and the mean hygroscopicity are important: the acquisition of water-soluble coatings on BC cores is expected to decrease the critical supersaturation. In addition to the LEO-fit technique, we also applied the delay time method, described in

Sect. 2.2.2, to investigate the influence of BC mixing state using SP2 data. This method makes it possible to

split BC-containing particles with a certain core size into two distinct classes, one containing exclusively "thickly" coated BC particles, the other one containing BC particles with "thin to moderate" coatings, with a classification threshold at a BC volume fraction of ~30 %. This binary mixing state analysis was applied to the BC data from three example stable cloud periods covering the range of $SS_{peak}$ typically encountered at the Jungfraujoch. Figure 9c shows the number fractions of particles in these two classes as a function of BC core diameter: the majority of BC-containing particles are thinly or moderately coated. It is important to note that this is not necessarily in contradiction to the expectation that BC at the Jungfraujoch is mostly internally mixed because the threshold between these two classes represents BC volume fractions as low as ~30 % (i.e. a coating thickness of 40-50 nm on a core with $D_{rBC}$ = 185 nm). Figure 9d shows the BC activated fraction curves segregated into the two coating thickness classes for the three cloud periods shown in Figure 9c. No coating effect on cloud droplet activation was found for the cloud with $SS_{peak}$ of 0.86 % (yellow lines). This indicates that at high supersaturation, essentially all BC-containing particles within the core size range covered by the SP2 activated to cloud droplets, even without substantial coating. By contrast, a distinct coating effect was found at lower $SS_{peak}$ when the activated fraction of smaller BC cores drops below the plateau values reached at larger cores. For the 5 July and 4 August periods, the activated fractions for the thickly coated class are always greater than or equal to the activated fractions for the thin-to-moderately coated class. This clearly shows that acquisition of coating increases the ability of a BC-containing particle with a certain BC core size to form a cloud droplet. This result qualitatively confirms the expectation from the Köhler theory that coating acquisition reduces the critical supersaturation for droplet activation of BC-containing particles through the combined effects of particle hygroscopicity (Raoult's law) and size (Kelvin effect). The relative influence of each of these two effects can however not be distinguished here, because a particle classified as thickly coated is both larger and more hygroscopic than another classified as thinly-to-moderately coated, for a fixed BC core size. It needs to be noted that the observed effect on the integrated BC mass scavenged fraction is small for the BC properties and clouds encountered at the Jungfraujoch, because the core size range in which activation is sensitive to the BC mixing state is clearly below the peak of the BC core mass size distribution (Fig. 9a) in most cases.

The size-segregated activation of BC cores observed in a previous study during a fog event at an urban site in Zurich, Switzerland, is also shown in Figure 9 (Motos et al., 2019). The peak supersaturation in this fog event were in the range 0.040-0.051 %, which is typical for mid-latitude fog (Hammer et al., 2014b) and almost an order of magnitude lower than the supersaturations in most of the clouds at the Jungfraujoch site. Accordingly, the activation onset diameter above which BC cores are activated to fog droplets is much greater, i.e. as large as ~130 nm for thickly coated BC cores and above around 230 nm for thinly-to-moderately coated BC. This explains the low BC scavenged fraction at the low fog supersaturations shown in Figure 8.

Ching et al. (2018) used the particle-resolved aerosol model PartMC-MOSAIC to simulate the aging of BC-containing particles in urban plumes. They modeled two-dimensional BC cores size and mixing state distributions, then inferred size-segregated activation curves and integrated scavenged fractions for BC using the $\kappa$-Köhler theory. It is not possible to directly compare the results of their model simulations with our experimental observations because of potential differences in BC size distributions and mixing states between the different environments. Nevertheless, our results confirm several of their findings in a qualitative manner. First, the simulated and observed activated fractions as a function of BC core diameter (e.g. the curves shown in Figure 9a) are of very similar shape and exhibit the same dependence on $SS_{peak}$. Second, our observation that

SS$_{peak}$ is the main parameter controlling the BC scavenged fraction, and that the sensitivity to the BC mixing state increases with decreasing supersaturation, was reproduced in their simulations.

The delay time method, which was applied to the SP2 data for the analyses presented above, only provides a binary mixing state classification. Quantitative mixing state information can be retrieved from the SP2 data using the LEO-fit approach described in Sect. 2.2.2, though at the expense of limiting the accessible size range. Coating thickness distributions for BC cores with mass equivalent diameters in the range 170 nm $\leq D_{rBC} \leq$ 200 nm are shown in Figure 10a for unactivated particles (interstitial inlet), droplet residual particles (PCVI inlet) and all particles (total inlet) sampled for a short period during the 4 August stable cloud period. Results are only available for this short period due to problems with the PCVI inlet discussed in Sect. 3.7. Fortunately, supersaturations were low during this cloud period (0.20 %; Table 1), which meant some BC stayed in the interstitial aerosol (Fig. 8). Figure 10a shows that, on average, the droplet residual BC-containing particles were more thickly coated than the interstitial BC-containing particles. Corresponding activated fractions as a function of coating thickness are shown in Figure 10b, segregated by the inlet pairs used to perform the calculation with either Eq. 1 or 2. A robust trend of gradually increasing activated fraction with increasing coating thickness was observed despite considerable uncertainty of these data points due to limited counting statistics and comparing measurements that were taken with minor difference (switching valves approach). This shows that acquisition of coatings facilitates activation, consistently with the delay time method results shown in Figure 9d.

## 4.4 Closure between predicted and observed cloud droplet activation of BC

The results presented in the previous section demonstrate qualitatively that the acquisition of coatings by BC particles increases their ability to form cloud droplets. Here we go a step further by comparing calculated and observed droplet activation thresholds on a single BC-containing particle level. Combining the SP2 data with the CCN measurements makes it possible to predict the critical supersaturation of individual BC-containing particles as described in Sect. 3.6. This prediction can then be compared with the actual activation to cloud droplets, which is inferred from the SP2 measurements behind the total and interstitial inlets. Motos et al. (2019) performed such a closure study for the activation of BC in fog. Here we investigate activation of BC in clouds at the Jungfraujoch, which typically have much higher peak supersaturations than fog.

Results for two example cloud periods are shown in Figure 11. The properties of individual BC-containing particles are shown in Panels a1 and b1 for the samples taken behind the total inlet (grey data points) and the interstitial inlet (coloured by coating thickness). The cloud peak supersaturations are shown as light blue horizontal lines. In an ideal case, i.e. perfectly well defined and homogeneous peak supersaturation and aerosol composition, and negligible measurement noise and bias, one would expect that no interstitial particles show up below the light blue lines (i.e. activated fraction = 100 % since SS$_{crit}$ < SS$_{peak}$), and that the numbers of BC-containing particles behind the interstitial and total inlets are equal above the light blue lines (i.e. activated fraction = 0 %). A trend in this direction can be seen in the Panels 1 of Figures 11a and 11b. Aggregating the data and calculating activation curves as a function of predicted critical supersaturation (black lines in Figures 11a2 and 11b2) reveals that, within experimental uncertainty, the supersaturation corresponding to the half-activation threshold for BC-containing particles indeed coincides with peak supersaturation for both cloud events. This demonstrates that closure between predicted and observed activation of the BC-containing particles

is achieved with the combination of the $\kappa$-Köhler theory, the ZSR mixing rule, and SP2 measurements of particle properties.

An alternative but equivalent method of performing this closure exercise is to compare the observed activation spectrum of the BC-containing particles (black lines in Figures 11a2 and 11b2) with that of the total aerosol based on SMPS measurements (green lines; inferred from the size-segregated activation spectra shown in Figure 7). The activated fraction of the total aerosol reaches $D_{half}^{cloud}$ exactly at the cloud peak supersaturation, because this is nothing less than the approach used to infer the peak supersaturation. The activation curves of BC-containing particles closely follow those of the total aerosol, indicating successful closure. The simple combination of the spherical core-shell morphology assumption, $\kappa$-Köhler theory, and the ZSR rule is a sufficiently detailed approach to predict how insoluble BC alters the droplet activation behaviour of BC-containing particles compared to that of BC-free particles.

The two stable cloud periods shown in Figure 11 have $SS_{peak}$ at the low end of the range of typical supersaturations observed at the Jungfraujoch. They were chosen because the BC-containing particle size range accessible to the SP2 LEO-fit approach is limited and some fraction of analyzable particles remaining in the interstitial aerosol was required to perform the analysis. The influence of BC mixing state on cloud droplet activation at different supersaturations was further assessed independently of particle size by calculating the activated fractions as a function of the coating thickness for BC-containing particles with a fixed overall optical diameter of $D_{opt} = 200$ nm. Results for 7 example cloud periods from both the CLACE2016 and CLACE2010 campaigns are shown in Figure 12. In clouds with high $SS_{peak}$, the activated fraction of BC-containing particles (coloured solid lines) equals that of BC-free particles of the same size (triangles attached to horizontal dashed lines), regardless of whether or not the BC-containing particles are coated. This is in agreement with the theoretically expected behaviour: the threshold coating thickness, calculated with the $\kappa$-Köhler theory and the ZSR mixing rule and indicated with a cross at half-rise activated fraction for each cloud period in Figure 12, is negligible at $SS_{peak} \geq \sim 0.5$ % for BC-containing particles with an overall diameter of 200 nm. During the cloud periods with low to medium $SS_{peak}$ in the range from 0.15 % to 0.3 %, the activation plateau at thick coatings is clearly less than 100 %. However, this is a result of temporal variations in $SS_{peak}$, and not due to properties of the BC-containing particles, since the activated fractions of BC-free particles with a diameter of 200 nm agree with the activation plateaus of BC-containing particles with thick coatings. The BC cores with no or very thin coatings do not activate at these low to medium $SS_{peak}$. Furthermore, the coating thickness at half-rise activated fraction (open circles in Figure 12) agrees with the theoretically expected threshold coating thickness within measurement uncertainty. This successful closure between predicted and observed activation thresholds again confirms that under the simplifying assumptions applied in this study (spherical core-shell morphology, surface tension of pure water), the combination of the $\kappa$-Köhler theory and the ZSR mixing rule is a suitable method of predicting the activation of BC-containing particles of known size and mixing state.

It has to be noted that, in the present study, we have tested the validity of the $\kappa$-Köhler theory and the ZSR mixing rule for only a small subset of large BC cores with very thin coatings at medium to high $SS_{peak}$, due to the limited size range of particles accessible to the SP2 LEO-fit analysis. Motos et al. (2019) used the same approach in fog and showed that this simplified theoretical approach is also valid at very low supersaturations and for a wider range of BC core sizes. Therefore, the method still remains to be examined against field observations of small to medium BC core sizes at medium to high $SS_{peak}$ (e.g. $SS_{peak} > 0.3$ %). However, the

scavenged fraction of BC in atmospheric clouds is only weakly sensitive to the exact behaviour of small to medium BC cores in this supersaturation range (e.g. Fig. 8), such that potential errors in the simplified theoretical model would be of little consequence. The closure achieved in this study for atmospheric BC is in line with a recent controlled laboratory study using a similar theoretical model and SP2 measurements of BC size and mixing state (Dalirian et al., 2018).

## 5 Conclusions

Two field experiments with *in-situ* cloud measurements were performed at the high altitude research station Jungfraujoch, central Switzerland, in summer 2010 and 2016. We selectively sampled the interstitial aerosol (unactivated particles), cloud droplet residual particles, and the total aerosol (sum of interstitial and droplet residual particles) using three different inlets with the aim of investigating the influence of size and mixing state on the activation of BC-containing particles to droplets in ambient clouds. We showed that the cloud peak supersaturation is the main parameter controlling the BC mass scavenged fraction. Variations in BC core size distribution and BC mixing state also have a minor influence on the scavenged fraction, particularly at higher supersaturations. It was qualitatively shown that, as expected, acquisition of coating increases the ability of BC cores of a certain size to activate to cloud droplets. Furthermore, quantitative closure between predicted and observed threshold coating thicknesses was achieved. Successful closure for the activation of BC was also achieved in a previous study in fog with lower peak supersaturations (Motos et al., 2019). These findings validate the approach of combining $\kappa$-Köhler theory and the ZSR mixing rule, under the simplifying assumptions of spherical core-shell morphology and surface tension of pure water, to predict the droplet activation of BC-containing particles of known size and mixing state. This theoretical approach has recently been applied in regional- and global-scale simulations with mixing state-resolved aerosol schemes to more accurately simulate nucleation scavenging and the life cycle of BC (Matsui, 2016; Ching et al., 2018). The experimental and closure-model results achieved in this study support such model simulations and imply that simulated BC scavenged fractions are accurate to the degree that other controlling parameters such as BC core size distribution, BC mixing state and cloud peak supersaturation are correctly simulated.

**Data availability.** Data used in this article are available in . Reconstructed overall particle diameter and coating thickness data on a single particle level (using the LEO-fit analysis) are available upon request to the corresponding author.

**Supplement.** The supplement related to this article is available online at:

**Author contribution.** MG and UB acquired the funding. MG conceptualized the study and the experiment was designed with JS and GM. MG supervised the study together with JS and UB. GM, JS and NK performed the field campaign and JCC contributed to instrument preparation and maintenance. GM analyzed and validated the experimental data with support from RM, JCC, JS, MG, and NK. GM prepared the manuscript and all co-authors contributed to interpretation of the results as well as manuscript review and editing.

**Acknowledgements.** We thank Nicolas Bukowiecki and Erik Herrmann for their help during the CLACE2016 campaign, and Ernest Weingartner, Zsofia Jurányi and Emanuel Hammer for their contributions to the CLACE2010 campaign. We also thank the International Foundation High Altitude Research Station Jungfraujoch and Gornergrat (HSFJG) for giving us the opportunity to perform an intensive campaign in addition to the continuous measurements in the Sphynx laboratory of the Jungfraujoch. This work was supported by the ERC under grant ERC-CoG-615922-BLACARAT and the EU FP7 project BACCHUS (grant no. 603445). Part of the observations included in this work originate from the continuous aerosol measurements at the Jungfraujoch site, which are supported by MeteoSwiss in the framework of the Swiss contributions (GAW-CH and GAW-CH-Plus) to the Global Atmosphere Watch programme of the World Meteorological Organization (WMO), and also supported by the ACTRIS2 project (funded by the EU H2020-INFRAIA-2014-2015 grant agreement no. 654109 and by the Swiss State Secretariat for Education, Research and Innovation (SERI) under contract number 15.0159-1; the opinions expressed and arguments employed herein do not necessarily reflect the official views of the Swiss Government). Meteorological measurements from the SwissMetNet Network were obtained through MeteoSwiss.

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

**Table 1. Parameters for all 14 stable cloud periods further analyzed in this study (3 from CLACE2010 where names are associated with an asterisk, 11 from CLACE2016). $N_{90}$ is the number concentration of potential CCN, i.e. particles with a mobility diameter larger than 90 nm. The droplet number concentration is estimated based on the difference between the particle number concentrations measured behind the total and the interstitial inlets. $\kappa_{smooth}$ refers to the moving average the CCN-derived hygroscopicity parameter $\kappa$ time series (see Sect. 3.5). 1-σ uncertainties are indicated by the symbol "Δ". Values of $T_{cloud\ base}$ are the temperatures that were used for calculating $SS_{peak}$.**

| Stable cloud period | Duration [min] | Median LWC [g m$^{-3}$] | Median $N_{90}$ [cm$^{-3}$] | Median droplet concentration [cm$^{-3}$] | activation plateau [%] | $D_{half}^{cloud}$ [nm] | Δ ($D_{half}^{cloud}$) [%] | $\kappa_{smooth}$ [-] | Δ ($\kappa_{smooth}$) [%] | $SS_{peak}$ [%] | Δ ($SS_{peak}$) [%] | $T_{JFJ}$ [°C] (min/max) | $T_{cloud\ base}$ [°C] (min/max) |
|---|---|---|---|---|---|---|---|---|---|---|---|---|---|
| 16 June* | 100 | - | - | - | 95 | 100 | ± 10 % | 0.23 | ± 33 % | 0.30 | ± 32 % | -1.8/0.2 | Not calc. (0) |
| 22 July* | 140 | - | 611 | 348 | 96 | 150 | ± 18 % | 0.24 | ± 28 % | 0.15 | ± 40 % | 1.7/2.0 | Not calc. (0) |
| 28 July* | 120 | - | 149 | 190 | 98 | 53 | ± 8 % | 0.15 | ± 46 % | 0.96 | ± 38 % | -2.4/-1.3 | Not calc. (0) |
| 18-19 June | 36 | 0.11 | 14 | 79 | 84 | 58.3 | ± 24 % | 0.30 | ± 23 % | 0.62 | ± 37 % | -6.9/-6.5 | -6.5/-6.3 |
| 25 June | 36 | 0.16 | 528 | 523 | 84 | 135.3 | ± 16 % | 0.16 | ± 11 % | 0.23 | ± 28 % | 0.4/1.5 | 0.7/1.7 |
| 26-27a June | 54 | 0.51 | 197 | 195 | 95 | 112.1 | ± 9 % | 0.22 | ± 27 % | 0.26 | ± 24 % | -3.5/-2.8 | -2.5/-1.6 |
| 26-27b June | 36 | 0.14 | 129 | 200 | 95 | 68.5 | ± 22 % | 0.27 | ± 17 % | 0.51 | ± 31 % | -6.7/-5.7 | -6.3/-5.0 |
| 5 July | 36 | 0.25 | 486 | 499 | 96 | 77.8 | ± 9 % | 0.17 | ± 23 % | 0.50 | ± 18 % | -1.3/-0.1 | -0.6/0.4 |
| 8-9 July | 36 | 0.19 | 600 | 607 | 81 | 78.1 | ± 8 % | 0.16 | ± 16 % | 0.51 | ± 22 % | -0.1/0.8 | 1.1/1.3 |
| 2a August | 36 | 0.24 | 27 | 64 | 97 | 38.6 | ± 35 % | 0.47 | ± 37 % | 0.86 | ± 42 % | 0.7/1.0 | 1.2/1.3 |
| 2b August | 36 | 0.38 | 76 | 134 | 100 | 41.0 | ± 16 % | 0.57 | ± 25 % | 0.72 | ± 24 % | 0.2/0.6 | 1.0/1.4 |
| 4 August | 54 | 0.26 | 779 | 474 | 80 | 125.8 | ± 15 % | 0.23 | ± 17 % | 0.20 | ± 24 % | 2.2/2.8 | 2.6/3.3 |
| 5-6a August | 90 | 0.34 | 46 | 93 | 86 | 77.0 | ± 16 % | 0.29 | ± 19 % | 0.41 | ± 27 % | -5.0/-4.2 | -4.3/-3.2 |
| 5-6b August | 36 | 0.21 | 35 | 113 | 84 | 56.1 | ± 18 % | 0.27 | ± 35 % | 0.68 | ± 32 % | -5.2/-4.7 | -4.5/-4.0 |

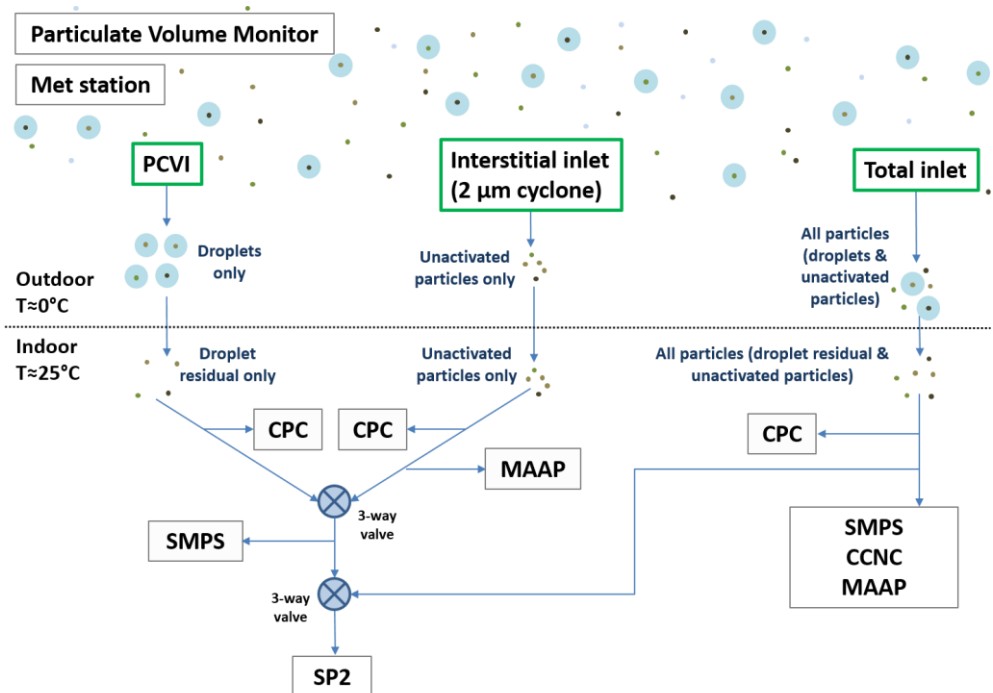

**Figure 1. Instrumental set-up.** Green rectangles indicate inlets, black rectangles instruments. Acronyms: PCVI: pumped counterflow virtual impactor, CPC: condensation particle counter, MAAP: multi-angle absorption photometer, SMPS: scanning mobility particle sizer, SP2: single particle soot photometer, CCNC: cloud condensation nuclei counter. Drying of the sample air occurs through the temperature increase from outdoor to indoor.

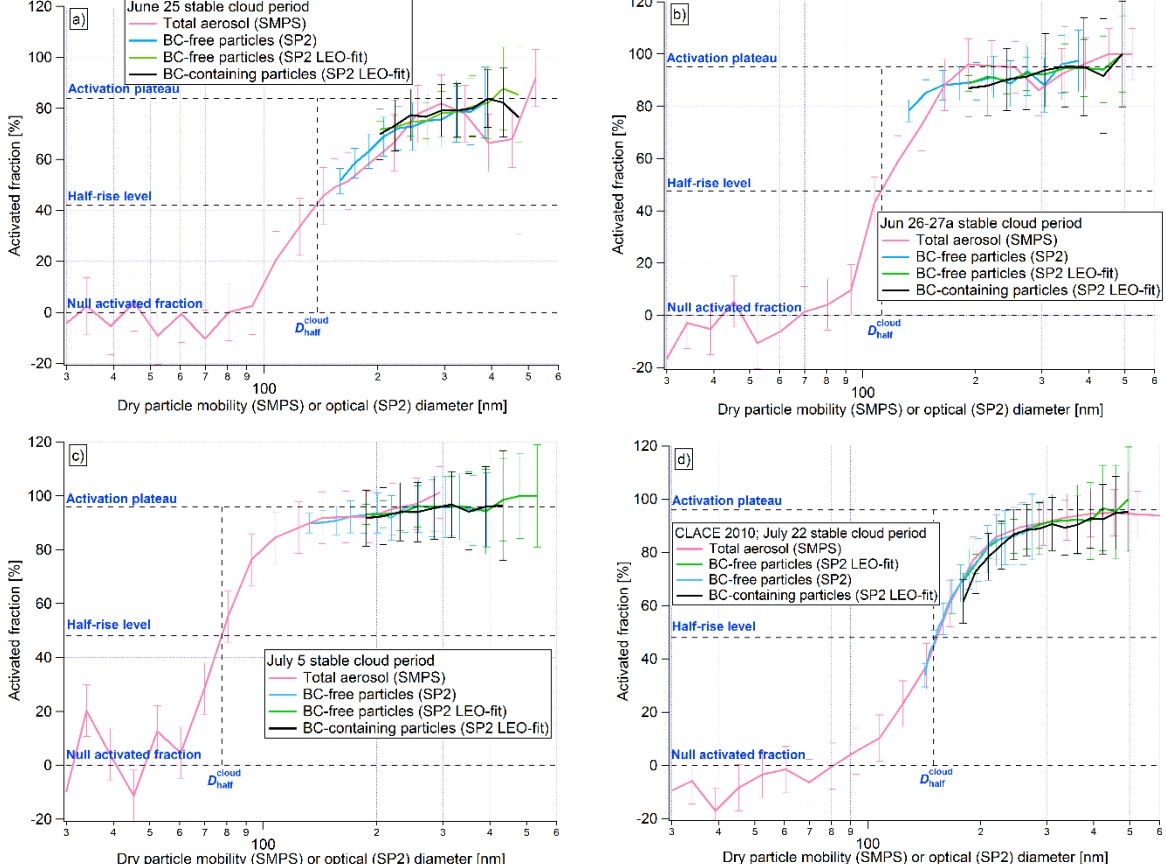

**Figure 2. Fraction of particles that activated to cloud droplets as a function of particle dry diameter as derived from the measurements behind the total and interstitial inlets for four example cloud events (averaged over the complete stable period). SMPS-derived activated fractions are shown against mobility diameter and include all particles, whereas SP2-derived data are separately shown for BC-free and BC-containing particles, both against optical diameter. BC-free particles are shown against optical diameter determined with standard optical sizing and against optical diameter determined using the LEO-fit approach in order to confirm consistency between the two. Each panel shows a different cloud event. The vertical dashed line marks the SMPS-derived half-rise threshold diameter for activation. Note that these activation spectra are averaged over a duration of 36 to 54 min, which may have resulted in a smearing of the activation transition if the cut-off diameter varied slightly in time.**

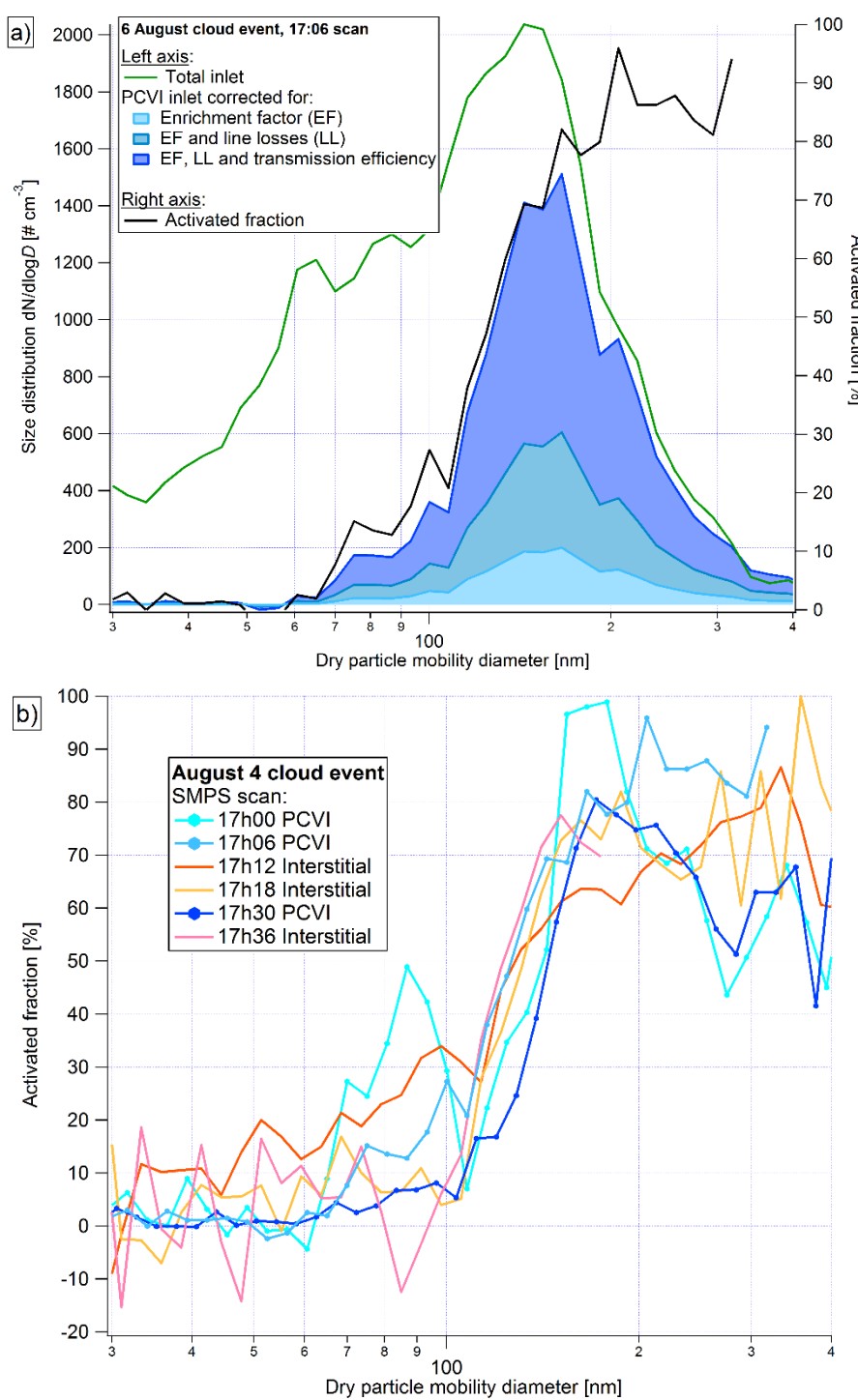

**Figure 3. Cloud droplet residual particle measurements using the PCVI inlet for the example of the 4 August cloud event (a) Particle number size distributions measured behind the total and PCVI inlets and corresponding activated fraction (Eq. 2). The corrections applied to the PCVI data are illustrated with the blue shadings. (b) Comparison between PCVI-derived and interstitial-derived activated particle fractions.**

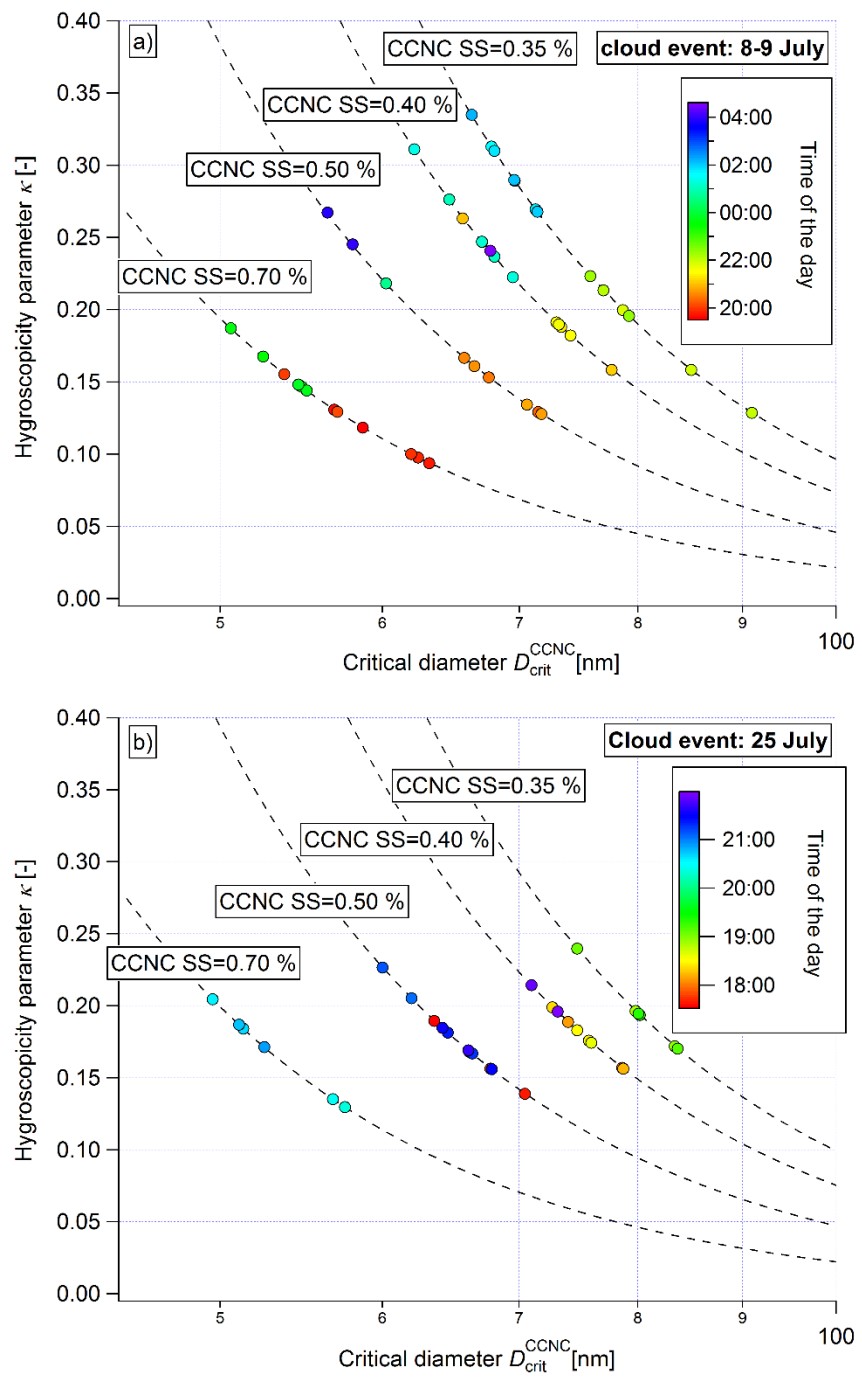

**Figure 4. Hygroscopicity parameter as a function of the critical diameter during the (a) 8-9 July and (b) 25 July cloud events. Single dots are data points extracted from a 6-min SMPS scan and simultaneous polydisperse CCNC measurement behind the total inlet. The thin dashed lines indicate the theoretical relationships between hygroscopicity and size for the supersaturation applied in the CCNC.**

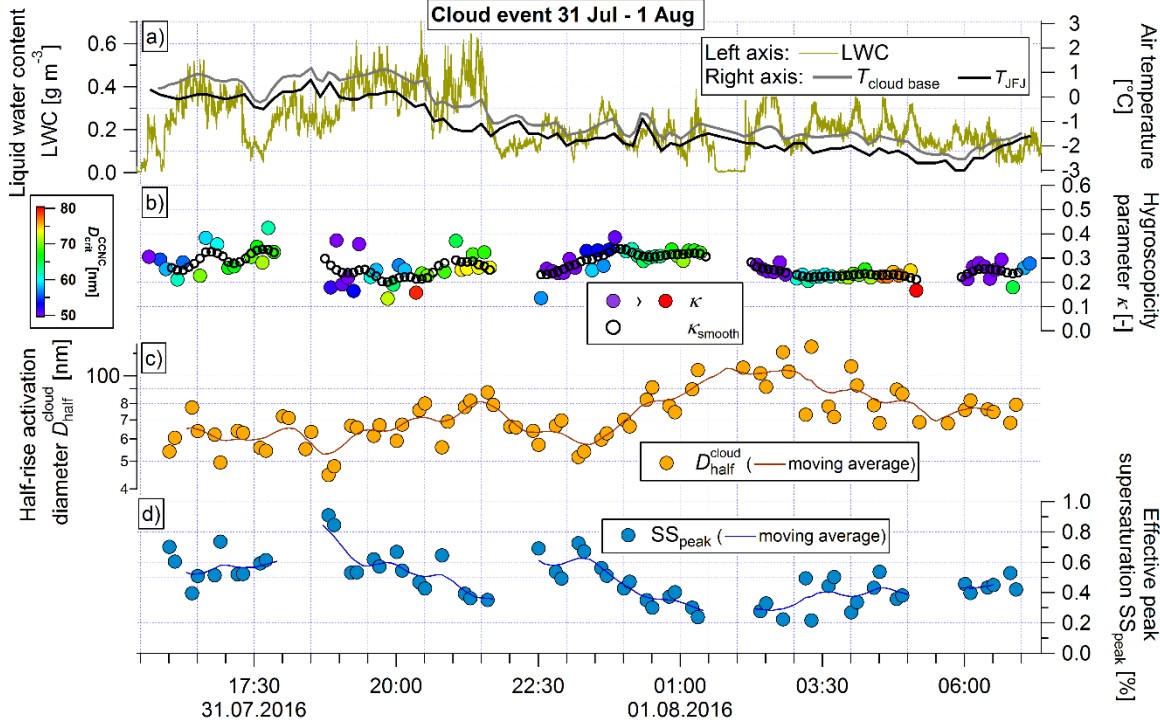

**Figure 5. 31 July-1 August full cloud event under northwestern wind conditions (a)** Time series of LWC, $T_{cloud\ base}$ and $T_{JFJ}$, **(b)** hygroscopicity parameter $\kappa$ as derived from polydisperse CCNC and SMPS measurements behind the total inlet. Values of $\kappa_{smooth}$, determined using the moving average of the time series of $\kappa$, are also shown. **(c)** Half-rise threshold dry diameter for cloud droplet activation $D_{half}^{cloud}$ and **(d)** cloud effective peak supersaturation $SS_{peak}$.

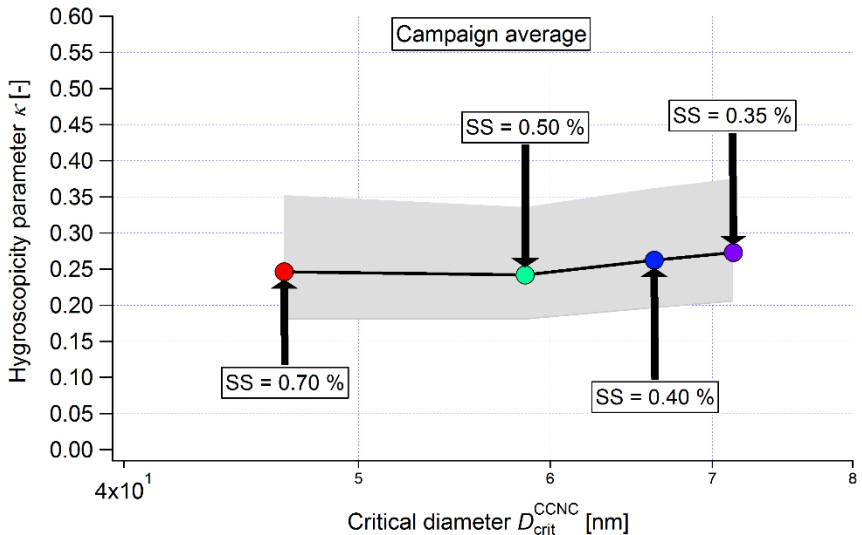

**Figure 6.** Hygroscopicity parameter $\kappa$ as a function of critical diameter retrieved from CCNC and SMPS measurements behind the total inlet. Data points represent values averaged over the whole campaign with a different colour for each supersaturation set in the CCNC. The grey shadings indicate the 25th and 75th percentiles of the $\kappa$ value. An example of underlying time-resolved data is shown in Figure 4.

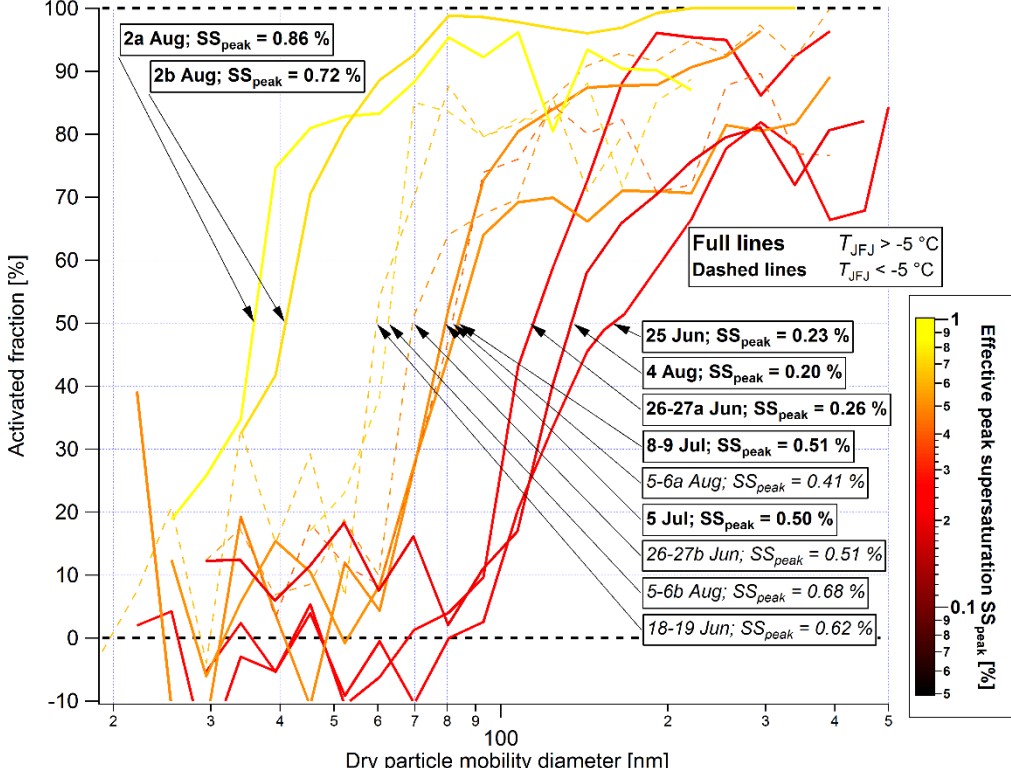

**Figure 7. Fraction of particles activated to cloud droplets for each stable cloud period of the CLACE 2016 campaign as derived from particle number size distributions measured by the SMPS behind the total and interstitial inlets. The lines are coloured by $SS_{peak}$ inferred from $D_{half}^{cloud}$ and $\kappa_{smooth}$ during the corresponding period. Periods of mixed-phase clouds are symbolized by dashed lines. For image clarity, error bars are not shown on this graph, but can be found in Figure 2 for three of these cloud periods.**

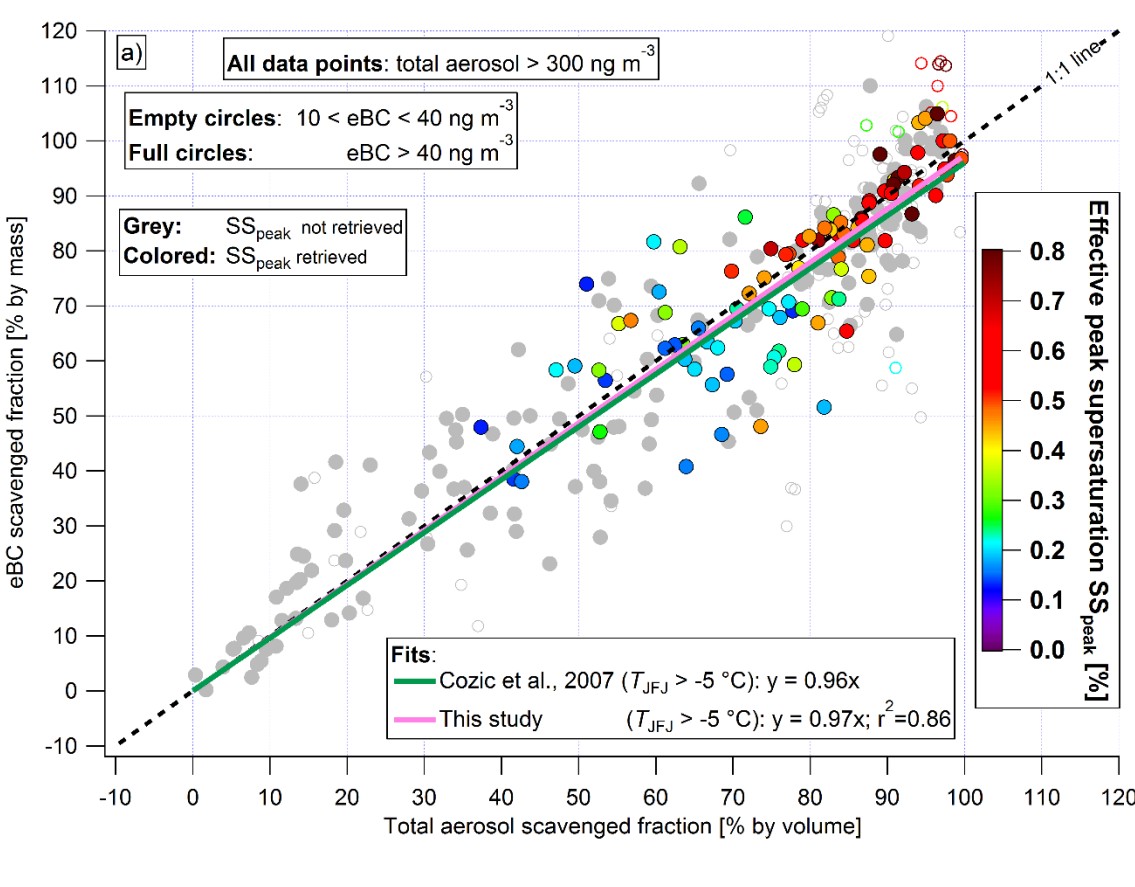

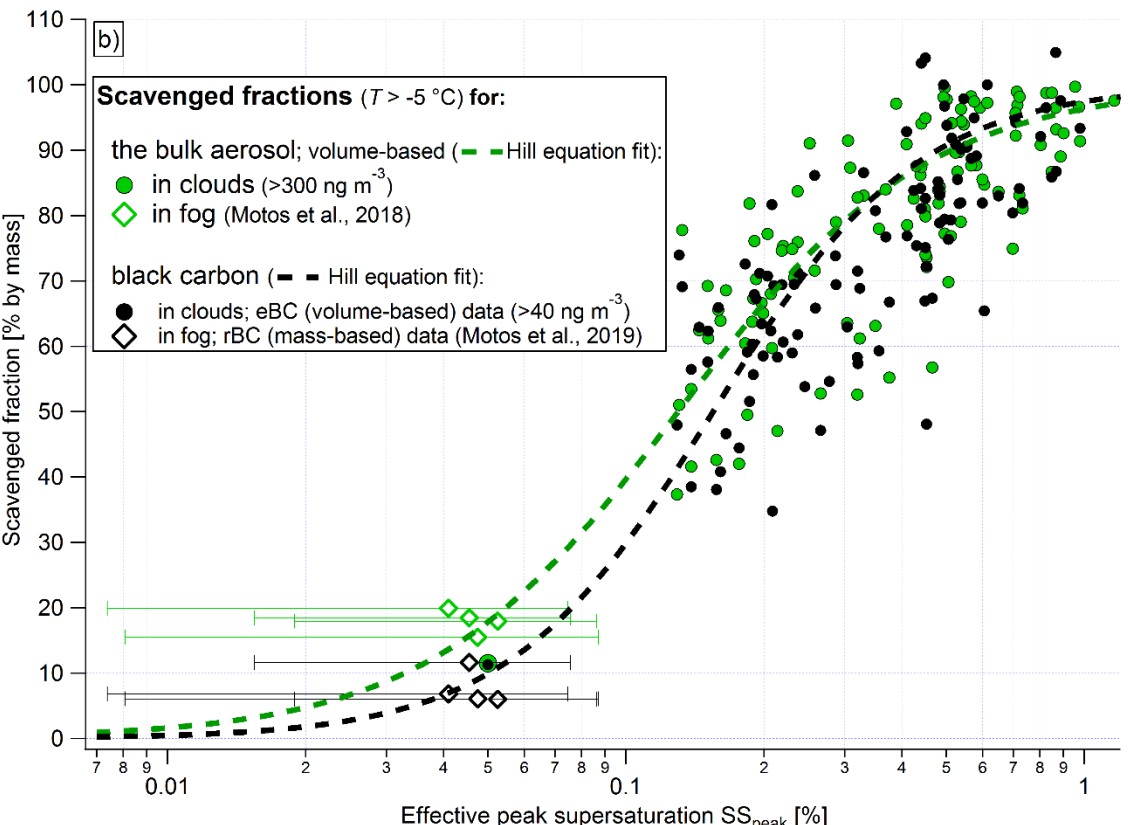

**Figure 8. eBC mass scavenged fractions derived from the two MAAPs and aerosol volume scavenged fractions derived from the two SMPSs during all liquid clouds. (a) Correlation of these two scavenged fractions with cloud effective peak supersaturation indicated by the colour, when available. (b) Dependence of scavenged fraction on cloud effective peak supersaturation. This panel additionally contains equivalent data from measurements in fog at an**

**urban site** (Motos et al., 2019)**. The coefficients of the manually fitted Hill equations (dashed lines) are provided in the Supplement. Open symbols are used in both panels for data points with concentrations close to the detection limits of the MAAP or SMPS (BC mass in the range 10-40 ng m$^{-3}$ or total volume ≤ 200 μm$^3$ cm$^{-3}$). Both panels are based on liquid clouds only. Potential mixed-phase clouds were filtered by excluding all data at temperatures below -5 °C.**

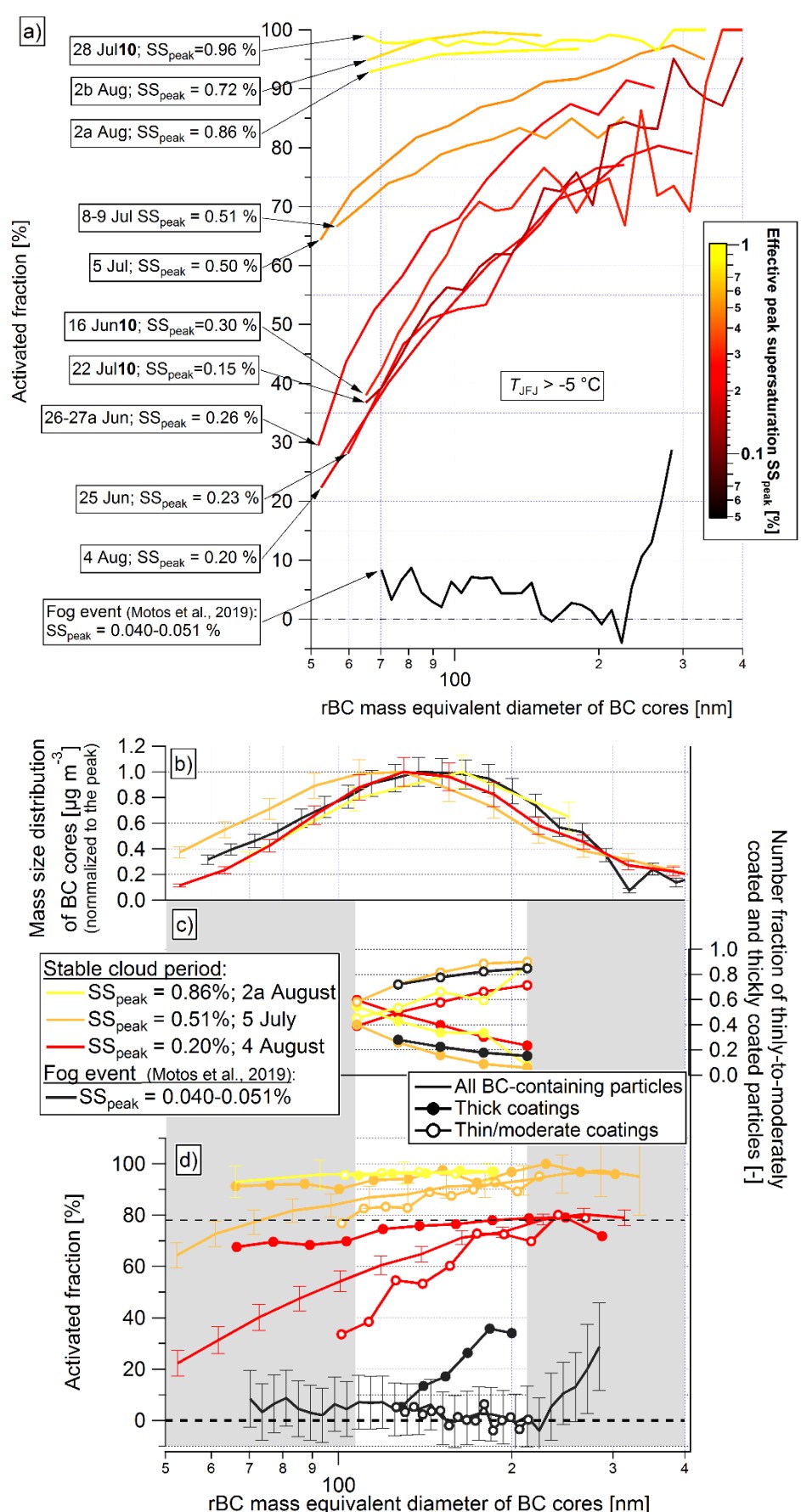

**Figure 9. Activation of BC to cloud droplets. (a) Activated fraction of all BC cores as a function of rBC mass equivalent diameter for all stable cloud periods with $T_{JFJ}$ > -5 °C, coloured by SS$_{peak}$. One fog event, extracted from**

the study of Motos et al. (2019), is also shown. Two variants of this figure, including clouds with $T_{JFJ}$ below -5 °C (Fig. Error! Reference source not found.**a) and SMPS-derived activation plateau (Fig.** Error! Reference source not found.**b) are shown in the Supplement. (b) BC core mass size distribution behind the total inlet, (c) number fraction of BC cores with thin to moderate or thick coatings as determined with the delay time method behind the total inlet. The most thickly coated particles, which caused saturation of the scattering signal, were included in the subset of BC-containing particles with thick coatings. The grey shadings indicate ranges of mass equivalent diameter for which the detection limits of the SP2 may introduce biases. (d) Activated fraction of all BC cores (same as in panel (a)) differentiated for the subsets of cores with thin to moderate coatings or thick coatings. This is only shown for 3 representative example stable cloud periods and one fog event.**

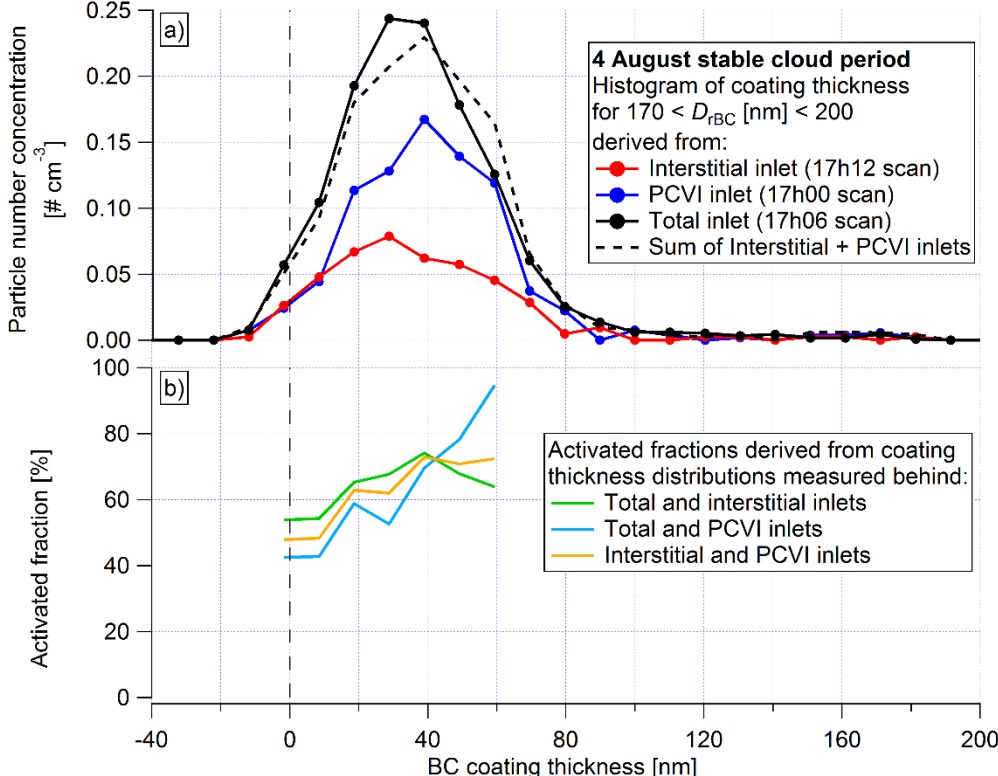

**Figure 10. (a) Histograms of coating thickness for BC cores with rBC core mass equivalent diameter between 170 and 200 nm as inferred from the SP2 data for the 4 August stable cloud period. (b) Activated fractions derived from the histograms shown in panel (a) and shown for the range of coating thicknesses between 0 and 60 nm, for which counting statistics are sufficient. The three calculations of activated fractions are redundant (based on data behind three inlets), this explains why the orange line is an average of the light blue and the green lines. Note that "unphysical" negative coating thickness values are caused by random noise for BC-containing particles with no or very little coating.**

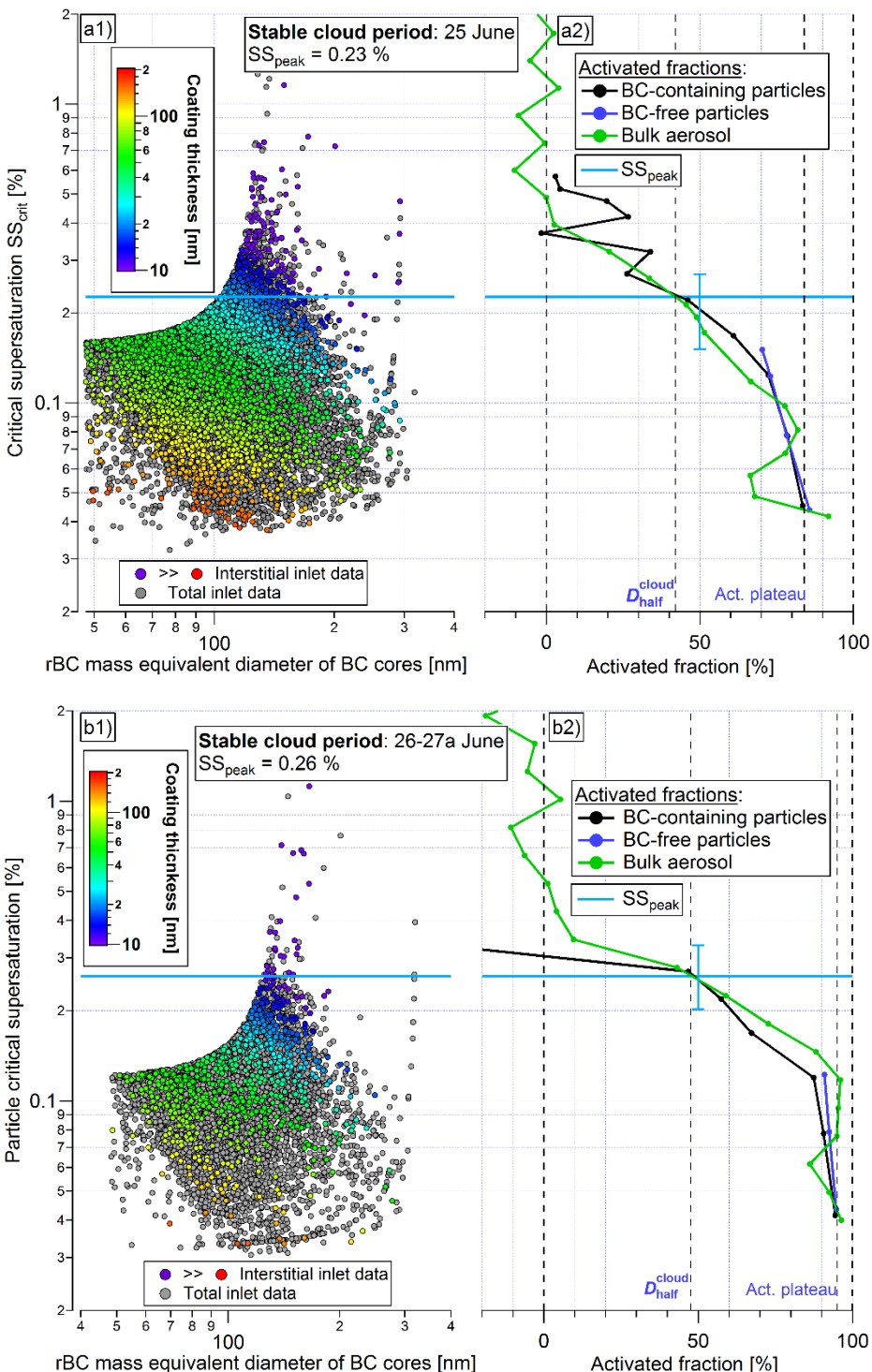

**Figure 11. Activation of BC-containing and BC-free particles during the 25 June (a) and 26-27a June (b) stable cloud periods. Panels (a1) and (b1) show the critical supersaturation of individual BC-containing particles as a function of rBC mass equivalent diameter coloured by coating thickness for the interstitial inlet and in grey for the total inlet. Note that the coating thickness can only be determined for BC-containing particles with an overall diameter greater than 180 nm, which explains the missing data points in the top left part of panels (a1) and (b1). Panels (a2) and (b2) depict the corresponding activated fraction of BC-containing particles as well as that of BC-free particles (SP2-derived) and bulk aerosol (SMPS-derived). Only one fourth of data points are shown in panel (a1) in order to visualize the fraction of points originating from interstitial inlet data compared to points from the total inlet data. Horizontal light blue lines indicate the value of $SS_{peak}$ retrieved using $D_{half}^{cloud}$ (method explained in Sect. 3.5). Note that values of $SS_{peak}$ for both cases (a) and (b) are at a level above which the SP2 detects only almost bare BC because small cores with substantial coating are outside the detection limits of the SP2.**

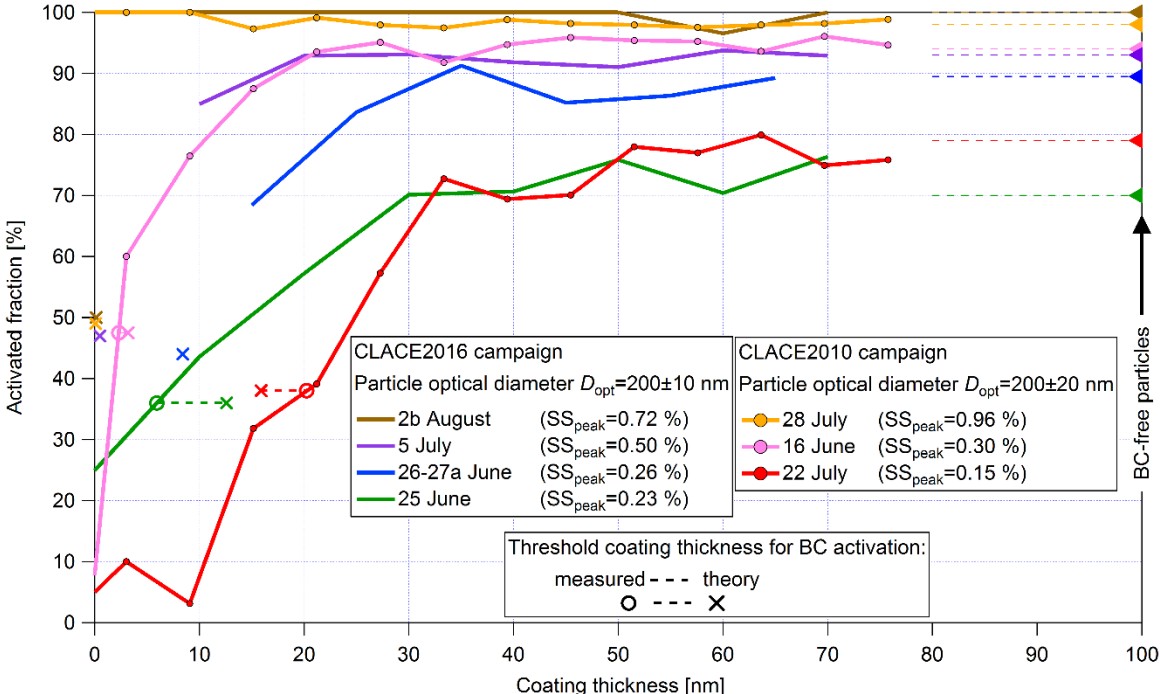

**Figure 12. Activated fractions as a function of coating thickness for BC-containing particles with an optical diameter of 200 nm during four stable cloud periods of the CLACE2016 campaign and three periods of the CLACE2010 campaign. Triangles accompanied by horizontal dashed lines correspond to the activated fraction of 200 nm BC-free particles, derived from Figure 2 (four example figures shown). The triangles are plotted at 100 nm coating thickness because this corresponds to 200 nm optical diameter if no BC core is present (BC-free particle). Dashed lines attached to activated fraction lines indicate the difference between experimentally observed (open circles) and theoretically predicted (crosses) coating thicknesses required for 200 nm BC-containing particles to reach activation up to half of the activation plateau.**