# Peer review of "Cloud droplet activation properties and scavenged fraction of black carbon in liquid-phase clouds at the high-alpine research station Jungfraujoch (3580 m a.s.l.)"

_Atmospheric Chemistry and Physics, 2018_

## Referee Comment (RC1) · Anonymous Referee #1 · 5 Dec 2018

The present manuscript describes the use of a sophisticated aerosol sampling system at a mountain site to investigate cloud droplet activation of both BC-free and BC-containing particles. The authors present unique data set of the in-situ CCN properties and scavenged fractions of these aerosols on a number and mass basis, which are useful for elucidating the parameters (water vapor supersaturation and aerosol microphysical properties) that control activation. Through the data analysis, they demonstrate that the simple k-Köhler theory under the assumptions of spherical core-shell structure of BC-containing particles can reasonably predict their cloud droplet activa-

tion, despite their complex morphology and non-ideal properties of coating materials in the real world. This is an important observational evidence that justifies the simple treatment of BC activation in regional and global models. Although I think some points in the manuscript need clarification, but overall I recommend this manuscript for publication in ACP after minor modification.

Specific comments:

P12, Line 30–31 and P13, Line 6–8: In these parts, the authors state that "Such variations in SS_peak are driven by variations in atmospheric dynamics (i.e., updraft)..." and "variations in D_half were mainly driven by variations in updraft velocities and resulting supersaturations,...", respectively. However, in addition to the updraft velocity, the SS_peak can also depend on total aerosol number concentrations (i.e., higher number concentrations of aerosols can lead to lower SS_peak). Therefore, the relationship between the absolute number concentrations of aerosols (total inlet) and SS_peak should be mentioned somewhere in the paper and the aerosol concentration data can be included in Table 1. Furthermore, to characterize cloud properties discussed in this paper, please consider including the LWC data in Table 1. The difference of the SMPS data between total and interstitial inlets may indicate the cloud droplet number concentrations, which might be also useful for characterizing cloud properties.

P14, Line 18–22: If the deviations of several data points in Figure 8a from 1:1 line are greater than measurement uncertainty, some data would indicate that BC particles were scavenged more efficiently than total aerosols. I do not understand the reason for that.

P15, Line 10–14: What refractive index values are assumed for BC core and BC-free aerosols for the SP2 data analysis?

Figure 9b and 9c: Which inlet for these SP2 data? Total inlet?

Figure 9c: Is the y-axis number fraction of thickly coated BC? The caption is not clear.

P16, Line 6–8: As the authors mentioned in P16, Line 1–4, the SS_peak values may be underestimated due to increased interstitial aerosol number concentrations due to WBF process in mixed phase clouds. Therefore, comparing the activated fraction for T < -5°C case and warm cloud case with "comparable SS_peak" looks logically inconsistent.

Figure 11: In this figure, coating thickness is indicated by color scale for rBC with mass equivalent diameter of about 50–300 nm. However, if the small BC (D_rBC ∼50 nm) has thin coatings (i.e., total optical diameter < 180 nm), the SP2 cannot quantify the coating thickness?

P19, Line 28, "lower" should be "higher"?

Minor corrections:

P3, Line 35: "(2016))" should be "(2016)".

P5, Line 20: "(Very...(2000)" should be "(Very...(2000))".

P17, Line 22: "DrBC" should be "D_rBC"

---

## Referee Comment (RC2) · Anonymous Referee #2 · 19 Dec 2018

The paper presents an experimental study on the cloud droplet activation of BC containing particles. The study was carried out at the high altitude Jungfraujoch station by sampling interstitial aerosol, cloud droplet residual particles and total aerosols during cloud periods. The authors used this dataset to investigate the influence of particle size and mixing state on the activation of BC containing particles. They also demonstrated that a simplified parameterization considering the core-shell structure of BC can be used in advanced modeling studies to describe the activation of BC into cloud droplets. The topic discussed in the paper is of general interest to the scientific community with

important applications in the climate and air quality. The paper is clear and well written. The experimental study, data analysis and data interpretation was carefully conducted. I suggest publication with only minor comments.

P2 L24: The authors state that "the coating focuses the solar radiation towards the BC core, enhancing its absorption, known as the lensing effect". The coating can have an opposite effect on BC absorption properties. Many studies have noticed that the light can be blocked by the coating as the coating absorption increases (e.g. Luo et al., 2018). Therefore the total absorption of BC containing particles can be weakened for BC with thickly coated by absorbing materials.

P6 L11-23: You mention two different approaches (i.e. LEO and delay time) to determine BC mixing state. Why did not you only use LEO fit to derive the BC coating thickness ? Please provide also the range of core thickness for your BC classifications (thickly, thin and moderate coatings).

P12 L 13: The kappa calculated for the bulk aerosol was found to be independent of the particle size. This is an important result as it suggest that a unique kappa can be used to predict CCN concentrations relevant to cloud droplet formation in modeling studies. How do you explain this result ? Is it specific to the Jungfraujoch site in summer ? Is it relevant for other seasons ? Was it already observed in other remote sites ?

P13 L7-11: "It does imply that variations in Dhalfcloud were mainly driven by variations in updraft velocities and resulting supersaturation, whereas differences in aerosol hygroscopicity only cause minor additional modulation of Dhalfcloud". I don't think that such conclusion can be derived from Figure 7. This figure only shows that the activation diameter increases for decreasing SS, which is not surprising. In my view there is missing discussion about key aerosol parameters that could play a important role on cloud properties. Did you look at the total particle number concentration and the characteristic size distributions, and in particular the contribution of accumulation mode particles ?

P13 L25: The authors attribute the relatively high scavenged fractions of the BC in the present study to the long-range transport that resulted in highly aged BC. BC particles can also obtain coating by in-cloud processes, such as aqueous-phase chemistry. Did you observe change in BC mixing state due to cloud processing by comparing SP2 measurements from the different inlets ? The authors could also look at BC coating before, during and after cloud events ? Moreover there is no information in the paper about air mass origins, their time of transport and the possible contribution of lower altitude pollution source, which could bring fresh BC particles especially in summertime. Did you observe seasonal variability of BC size and mixing state ? This is an important concern to assess the relevance of the results presented in the paper and the applicability of the derived parameterization.

P13 L38 – P14 L2: As the scavenged fraction of BC increase with SS, can we conclude from Fig 8 that nucleation scavenging of BC dominate over impaction scavenging with cloud droplets at Jungfraujoch ?

P15 L7-13 : Which refractive index did you use to retrieve the SP2 size distribution for BC-free particles ?

P16 L 26-40: Can we also conclude from Figure 9d that the activated fraction mainly depends on the BC total size (core+coating) while the chemical composition (hygroscopicity) of the coating appears to be of a secondary importance ?

Luo, J., Zhang, Y., Wang, F., and Zhang, Q.: Effects of brown coatings on the absorption enhancement of black carbon: a numerical investigation, Atmos. Chem. Phys., 18, 16897-16914, https://doi.org/10.5194/acp-18-16897-2018, 2018.

---

## Author Comment (AC1) · 14 Feb 2019

We thank the two referees for their constructive comments, definitely helpful to clarify certain points of the paper.

The page and line numbers enumerated in this document refer to the corrected maunuscript (without track changes).

Answers of the authors to the interactive comment of Anonymous Referee #1 (Referee Comment 1)

Anonymous review of manuscript: General remarks

The present manuscript describes the use of a sophisticated aerosol sampling system at a mountain site to investigate cloud droplet activation of both BC-free and BC-containing particles. The authors present unique data set of the in-situ CCN properties and scavenged fractions of these aerosols on a number and mass basis, which are useful for elucidating the parameters (water vapor supersaturation and aerosol microphysical properties) that control activation. Through the data analysis, they demonstrate that the simple $\kappa$-Köhler theory under the assumptions of spherical core-shell structure of BC-containing particles can reasonably predict their cloud droplet activation, despite their complex morphology and non-ideal properties of coating materials in the real world. This is an important observational evidence that justifies the simple treatment of BC activation in regional and global models. Although I think some points in the manuscript need clarification, but overall I recommend this manuscript for publication in ACP after minor modification.

Specific comments from Referee #1:

P12, Line 30–31 and P13, Line 6–8: In these parts, the authors state that "Such variations in SS_peak are driven by variations in atmospheric dynamics (i.e., updraft)..." and "variations in D_half were mainly driven by variations in updraft velocities and resulting supersaturations,...", respectively. However, in addition to the updraft velocity, the SS_peak can also depend on total aerosol number concentrations (i.e., higher number concentrations of aerosols can lead to lower SS_peak). Therefore, the relationship between the absolute number concentrations of aerosols (total inlet) and SS_peak should be mentioned somewhere in the paper and the aerosol concentration data can be included in Table 1. Furthermore, to characterize cloud properties discussed in this paper, please consider including the LWC data in Table 1. The difference of the SMPS data between total and interstitial inlets may indicate the cloud droplet number concentrations, which might be also useful for characterizing cloud properties.

Remark: A very similar comment was posted by the Anonymous Referee #2.

Response: The relationship between potential CCN, cloud droplet number, updraft velocity and cloud peak supersaturation, i.e. cloud formation in CCN-limited or updraft limited regimes, was extensively studied at the Jungfraujoch site by Hoyle et al. (2016).

Changes: We modified the paragraph mentioned by the Referee (p.12, l.34) to: "Such variations in SSpeak are primarily driven by variations in atmospheric dynamics (i.e. updraft) at the cloud base and to a lesser extent by the number concentration of potential CCN, as demonstrated for the Jungfraujoch site by Hoyle et al. (2016)." We included the median liquid water content (LWC), median particle number concentration of potential CCN (i.e. particles larger than 90 nm in diameter) and the median droplet number concentration during each cloud event in Table 1. Two new references to Table 1 were added, p.12, l.37 and p.13, l.14: "Variations in effective cloud peak supersaturation are a priori unrelated to the cloud LWC (Fig. 5a and Table 1)", "The key parameters for each selected cloud period are summarized in Table 1, including the droplet number concentration inferred from the difference in particle number concentration between the total and the interstitial inlet, and the median number concentration of potential CCN, i.e. particles with a mobility diameter larger than 90 nm (N90; e.g. Hammer et al.; 2014a)."

P14, Line 18–22: If the deviations of several data points in Figure 8a from 1:1 line are greater than measurement uncertainty, some data would indicate that BC particles were scavenged more efficiently than total aerosols. I do not understand the reason for that.

Response: This is an important observation, which we did not address. We have therefore introduced the following changes in the manuscript:

Changes: The last sentence of the following paragraph (p.14, l.25) has been added in the revised manuscript: "The scavenged fraction of BC mass is only expected to be equal to the total aerosol volume scavenged fraction for all peak supersaturations, if BC

contributes an equal fraction to the aerosol volume at any particle size and if the critical activation diameters of the BC-containing particles and total aerosol are equal. While the latter condition is closely fulfilled if BC is internally mixed with substantial coatings, size-independent BC volume fractions are a priori not expected. Nevertheless, the scavenged fractions of total aerosol volume and BC mass are essentially equal on average. However, deviations of several data points in Figure 8a from the "1:1"-line are greater than measurement uncertainty, indicating that even at remote locations the BC scavenged fraction can differ from the total aerosol volume scavenged fraction in individual cloud events, likely due to some size dependence of the contribution of BC to the aerosol volume and/or disagreement between the critical activation diameters of BC-free and BC-containing particles. For example new particle formation events followed by growth to sizes below the droplet activation cut-off diameter is one possible mechanism that can result in the BC scavenged fraction becoming greater than that of the total aerosol volume."

P15, Line 10–14: What refractive index values are assumed for BC core and BC-free aerosols for the SP2 data analysis?

Changes: We included the following paragraph into the experimental section (Sect. 2.2.2), p.6, l.17: "A refractive index of 1.50+0i was chosen to convert the scattering cross section measurements of BC-free particles to optical diameters, which brought the SP2 and SMPS derived size distributions in agreement in the overlapping size range (the optical sizing is only weakly sensitive to the choice of refractive index as shown by Taylor et al., 2015). For BC-containing particles, the same refractive index was used for the coatings, while 2.00+1.00i was chosen for the BC cores. This choice resulted in agreement between the optical diameters of the bare BC cores measured just before incandescence onset and the rBC mass equivalent diameters."

Figure 9b and 9c: Which inlet for these SP2 data? Total inlet?

Response: This is all behind the total inlet.

Changes: The caption of Figure 9 was adapted.

Figure 9c: Is the y-axis number fraction of thickly coated BC? The caption is not clear.

Changes: The legend and labels of the figure were modified to make it clear.

P16, Line 6–8: As the authors mentioned in P16, Line 1–4, the SS_peak values may be underestimated due to increased interstitial aerosol number concentrations due to WBF process in mixed phase clouds. Therefore, comparing the activated fraction for T < -5 C case and warm cloud case with "comparable SS_peak" looks logically inconsistent.

Changes: This paragraph now reads (p.16, l. 16): Mixed-phase or even completely glaciated clouds may occur at lower temperatures. Mixed-phase clouds may result in the conversion of particles from droplets (activated particles) to interstitial aerosol through the Wegener-Bergeron-Findeisen process (e.g. Cozic et al., 2007), thereby potentially obscuring the causal relationship between SSpeak and droplet activation. However, Verheggen et al. (2007) showed that D_halfˆcloud remains well-defined and that only small differences in average D_halfˆcloud exist between mixed-phase and liquid clouds. This suggests that the Wegener-Bergeron-Findeisen process does not affect the inferred SSpeak. The fact that the Wegener-Bergeron-Findeisen process evaporates some cloud droplets, whereby the droplet nuclei are released back to the interstitial aerosol, explains that the BC activated fraction was lower in most clouds at temperatures below 5 °C compared to that in warm clouds at comparable peak supersaturation (see the dashed lines in Figure S3a).

Figure 11: In this figure, coating thickness is indicated by color scale for rBC with mass equivalent diameter of about 50–300 nm. However, if the small BC (D_rBC 50 nm) has thin coatings (i.e., total optical diameter < 180 nm), the SP2 cannot quantify the coating thickness?

Response: Indeed, the LEO-fit technique cannot be applied to particles with an optical

diameter smaller than around 180 nm, whether they contain BC or not.

Changes: The text "particles larger than around 180 nm only" was added to the sentence "The reconstructed scattering amplitude is then used to infer the total particle optical diameter, for particles larger than around 180 nm only", in Sect. 2.2.2, p.6, l.14. The caption of Figure 11 was rewritten and now reads: Figure 11. Activation of BC-containing and BC-free particles during the 25 June (a) and 26-27a June (b) stable cloud periods. Panels (a1) and (b1) show the critical supersaturation of individual BC-containing particles as a function of rBC mass equivalent diameter coloured by coating thickness for the interstitial inlet and in grey for the total inlet. Note that the coating thickness can only be determined for BC-containing particles with an overall diameter greater than 180 nm, which explains the missing data points in the top left part of panels (a1) and (b1). Panels (a2) and (b2) depict the corresponding activated fraction of BC-containing particles as well as that of BC-free particles (SP2-derived) and bulk aerosol (SMPS-derived). Only one fourth of data points are shown in panel (a1) in order to visualize the fraction of points originating from interstitial inlet data compared to points from the total inlet data. Horizontal light blue lines indicate the value of SS-peak retrieved using $D\_half^{cloud}$ (method explained in Sect. 3.5). Note that values of SSpeak for both cases (a) and (b) are at a level above which the SP2 detects only almost bare BC because small cores with substantial coating are outside the detection limits of the SP2.

P19, Line 28, "lower" should be "higher"?

Changes: Manuscript corrected.

Minor corrections:

P3, Line 35: "(2016))" should be "(2016)".

P5, Line 20: "(Very...(2000)" should be "(Very...(2000))".

P17, Line 22: "DrBC" should be "D_rBC"

[Figure]

Changes: Thank you; Manuscript corrected.

References (for responses to both referees):

[revised manuscript text omitted]

---

## Author Comment (AC2) · 14 Feb 2019

We thank the two referees for their constructive comments, definitely helpful to clarify certain points of the paper.

The page and line numbers enumerated in this document refer to the corrected maunuscript (without track changes).

Answers of the authors to the interactive comment of Anonymous Referee #2 (Referee Comment 2)

[Figure]

Anonymous review of manuscript: General remarks

The paper presents an experimental study on the cloud droplet activation of BC containing particles. The study was carried out at the high altitude Jungfraujoch station by sampling interstitial aerosol, cloud droplet residual particles and total aerosols during cloud periods. The authors used this dataset to investigate the influence of particle size and mixing state on the activation of BC containing particles. They also demonstrated that a simplified parameterization considering the core-shell structure of BC can be used in advanced modeling studies to describe the activation of BC into cloud droplets. The topic discussed in the paper is of general interest to the scientific community with important applications in the climate and air quality. The paper is clear and well written. The experimental study, data analysis and data interpretation was carefully conducted. I suggest publication with only minor comments.

Specific comments from Referee #2:

P2 L24: The authors state that "the coating focuses the solar radiation towards the BC core, enhancing its absorption, known as the lensing effect". The coating can have an opposite effect on BC absorption properties. Many studies have noticed that the light can be blocked by the coating as the coating absorption increases (e.g. Luo et al., 2018). Therefore the total absorption of BC containing particles can be weakened for BC with thickly coated by absorbing materials.

Response: The absorption Ångström exponent of the aerosol at Jungfraujoch is typically close to that of BC, indicating that brown carbon and thus the above-mentioned effect is only of minor importance. Nevertheless, we included this possible effect in the general part of the introduction.

Changes: The corresponding paragraph, in Sect. 1, p.1, l.22, was modified: "[. . .] firstly, the coating modifies the particle absorption with effects that are still under debate. Some studies reported an absorption enhancement assuming that the coating focuses the solar radiation towards the BC core, this is known as the lensing effect (e.g. Fuller

et al., 1999; Bond et al., 2006). Other studies hypothesized that the coatings block the radiation, resulting in a reduction of the absorption by BC (e.g. Luo et al., 2018). Secondly,[. . .]"

P6 L11-23: You mention two different approaches (i.e. LEO and delay time) to determine BC mixing state. Why did not you only use LEO fit to derive the BC coating thickness ? Please provide also the range of core thickness for your BC classifications (thickly, thin and moderate coatings).

Response: As mentioned in the experimental section, these two methods have their advantages and their limitations: The LEO-fit is quantitative but restricted to a rather narrow range of optical particle diameters. The delay time method only gives a binary classification of BC mixing state with the advantage of covering a wider size range. Therefore we used both methods.

Changes: We modified the following chapter in Sect. 4.3, p.16, l.34: "In addition to the LEO-fit technique, we also applied the delay time method described in Sect. 2.2.2 to investigate the influence of BC mixing state using SP2 data. This method makes it possible to split BC-containing particles with a certain core size into two distinct classes, one containing exclusively "thickly" coated BC particles, the other one containing BC particles with "thin to moderate" coatings, with a classification threshold at a BC volume fraction of $\sim$70 %."

P12 L 13: The kappa calculated for the bulk aerosol was found to be independent of the particle size. This is an important result as it suggest that a unique kappa can be used to predict CCN concentrations relevant to cloud droplet formation in modeling studies. How do you explain this result ? Is it specific to the Jungfraujoch site in summer ? Is it relevant for other seasons ? Was it already observed in other remote sites ?

Response: The climatology of aerosol hygroscopicity is the central topic of the HTDMA-based study by Kammermann et al. (2010), which is referenced in the manuscript. The fact that the hygroscopicity is independent of particle size was one of their main results,

and it was confirmed for CCN-derived kappa at Jungfraujoch by Jurányi et al. (2011) and Hammer et al. (2014). The study by Schmale et al. (2018) provides the most comprehensive CCN, composition and size distribution data set at remote locations, which is being used to benchmark global model simulation (Fanourgakis et al., 2019). These studies are also referenced. The authors of this manuscript consider that there is sufficient dedicated literature on this topic and that there is no need to elaborate more on it in this manuscript.

Changes: No changes to the manuscript.

P13 L7-11: "It does imply that variations in D_half^cloud were mainly driven by variations in updraft velocities and resulting supersaturation, whereas differences in aerosol hy-groscopicity only cause minor additional modulation of Dhalfcloud". I don't think that such conclusion can be derived from Figure 7. This figure only shows that the activa-tion diameter increases for decreasing SS, which is not surprising. In my view there is missing discussion about key aerosol parameters that could play a important role on cloud properties. Did you look at the total particle number concentration and the char-acteristic size distributions, and in particular the contribution of accumulation mode particles ?

Remark: A very similar comment was posted by the Anonymous Referee #1.

Response: The relationship between potential CCN, cloud droplet number, updraft velocity and cloud peak supersaturation, i.e. cloud formation in CCN-limited or updraft limited regimes, was extensively studied at the Jungfraujoch site by Hoyle et al. (2016).

Changes: We modified the paragraph mentioned by the Referee (p.12, l. 34) to: "Such variations in SSpeak are primarily driven by variations in atmospheric dynamics (i.e. updraft) at the cloud base and to a lesser extent by the number concentration of potential CCN, as demonstrated for the Jungfraujoch site by Hoyle et al. (2016)." We included the median liquid water content (LWC), median particle number concentration of potential CCN (i.e. particles larger than 90 nm in diameter) and the median droplet

number concentration during each cloud event in Table 1. Two new references to Table 1 were added, p.12, l.37 and p.13, l.14: "Variations in effective cloud peak supersaturation are a priori unrelated to the cloud LWC (Fig. 5a and Table 1)" "The key parameters for each selected cloud period are summarized in Table 1, the droplet number concentration inferred from the difference in particle number concentration between the total and the interstitial inlet, and the median number concentration of potential CCN, i.e. particles with a mobility diameter larger than 90 nm (N90; e.g. Hammer et al.; 2014a)."

P13 L25: The authors attribute the relatively high scavenged fractions of the BC in the present study to the long-range transport that resulted in highly aged BC. BC particles can also obtain coating by in-cloud processes, such as aqueous-phase chemistry. Did you observe change in BC mixing state due to cloud processing by comparing SP2 measurements from the different inlets ? The authors could also look at BC coating before, during and after cloud events ? Moreover there is no information in the paper about air mass origins, their time of transport and the possible contribution of lower altitude pollution source, which could bring fresh BC particles especially in summertime. Did you observe seasonal variability of BC size and mixing state ? This is an important concern to assess the relevance of the results presented in the paper and the applicability of the derived parameterization.

Response: Two different topics are addressed in this comment:

1)On the reason for high BC scavenged fractions:

This was the conclusion of the study by Cozic et al. (2007) at the same site, as mentioned in the introduction of our manuscript (p.13, l.30): "They [Cozic et al., 2007] found close agreement between the scavenged fractions of BC and that of the total aerosol for warm clouds with temperature at Jungfraujoch (TJFJ) above -5 °C, i.e. high correlation and almost identical values on average. Such close agreement is a priori not expected because BC is insoluble in water; however, the authors attributed it to the high degree of internal mixing of BC in the aged aerosol at the Jungfraujoch."

However, our results contradict this conclusion and show that cloud supersaturation is the main driver of BC scavenging. BC aging/hygroscopicity is a secondary controlling factor. The fact that scavenged fractions are high at the JFJ is mainly explained by the high cloud supersaturations, which results in activation cut-off diameters being smaller than the peak of the mass size distributions at this site.

2)Relation of BC mixing state to air mass origin and aging processes

Changes: The text on p.15, l.32 was modified to: "Such agreement indicates that the majority of the BC-containing particles with a diameter greater than 180 nm consist of small BC cores with substantial coating acquired through various processes during atmospheric transport to the remote Jungfraujoch site (through e.g. condensation of volatile organic compounds, coagulation with particles or in-cloud processes)."

Currently, we are preparing another manuscript using SP2 data from two summer campaigns (including the summer 2016 data) and one winter campaign at the Jungfraujoch, focusing on the variability and seasonality of BC size and mixing state for boundary layer influenced and free troposphere conditions. Differences in coating thickness will be addressed in detail there.

With respect to effects of in-cloud processing on BC mixing state, unfortunately, the data set does not allow us draw any conclusions (we would be very happy to do so if possible).

P13 L38 – P14 L2: As the scavenged fraction of BC increase with SS, can we conclude from Fig 8 that nucleation scavenging of BC dominate over impaction scavenging with cloud droplets at Jungfraujoch ?

Response: Yes, this is an interesting point; the authors agree that this should be mentioned in the manuscript.

Changes: The following paragraph is added to Sect 4.2, p.14, l.13: "This result also confirms that nucleation scavenging is the dominant mechanism resulting in the incorporation of particles (BC-free or BC-containing) into cloud droplets at the Jungfraujoch. If impaction, a process unrelated to SSpeak, were to dominate, we would not observe such a relationship between the scavenged fractions and SSpeak."

P15 L7-13: Which refractive index did you use to retrieve the SP2 size distribution for BC-free particles ?

Response: The authors agree that this information should be available in the manuscript (also asked by Referee #1).

Changes: We included the following paragraph into the experimental section (Sect. 2.2.2, p.6, l.17): "A refractive index of 1.50+0i was chosen to convert the scattering cross section measurements of BC-free particles to optical diameters, which brought the SP2 and SMPS derived size distributions in agreement in the overlapping size range (the optical sizing is only weakly sensitive to the choice of refractive index as shown by Taylor et al. (2015). For BC-containing particles, the same refractive index was used for the coatings, while 2.00+1.00i was chosen for the BC cores. This choice resulted in agreement between the optical diameters of the bare BC cores measured just before incandescence onset and the rBC mass equivalent diameters."

P16 L 26-40: Can we also conclude from Figure 9d that the activated fraction mainly depends on the BC total size (core+coating) while the chemical composition (hygroscopicity) of the coating appears to be of a secondary importance ?

Response: The activated fractions shown in Figure 9d combine changes in particle size and hygroscopicity: for a fixed BC core size, a particle classified, based on the delay time method, as "thickly coated" is both larger in diameter and more hygroscopic than a particle classified as "thinly-to-moderately coated". It is therefore difficult to draw unambiguous conclusions on the relative importance of particle size versus mixing state based on this Figure. However, Figure 12 considers only particles with an optical diameter of 200 nm, it is thus possible to assess the importance of particle mixing state/hygroscopicity based on Figure 12 (see second change).

Changes: The following paragraph (Sect. 4.3, p.17, l.13) was modified and now reads: "This result qualitatively confirms the expectation from the Köhler theory that coating acquisition reduces the critical supersaturation for droplet activation of BC-containing particles through the combined effects of particle hygroscopicity (Raoult's law) and size (Kelvin effect). The relative influence of each of these two effects can however not be distinguished here, because a particle classified as thickly coated is both larger and more hygroscopic than another classified as thinly-to-moderately coated, for a fixed BC core size. It needs to be noted that the observed effect [. . .]"

We also added "independently of particle size" to the following paragraph (p.19, l.11): The influence of BC mixing state on cloud droplet activation at different supersaturations was further assessed independently of particle size by calculating the activated fractions as a function of the coating thickness for BC-containing particles with a fixed overall optical diameter of Dopt = 200 nm.

References (same for response to both referees):

Bond, T. C., Habib, G. and Bergstrom, R. W.: Limitations in the enhancement of visible light absorption due to mixing state, Journal of Geophysical Research, 111(D20), doi:10.1029/2006JD007315, 2006. Cozic, J., Verheggen, B., Mertes, S., Connolly, P., Bower, K., Petzold, A., Baltensperger, U. and Weingartner, E.: Scavenging of black carbon in mixed phase clouds at the high alpine site Jungfraujoch, Atmospheric Chemistry and Physics, 7(7), 1797–1807, 2007.

Fanourgakis, G. S., Kanakidou, M., Nenes, A., Bauer, S. E., Bergman, T., Carslaw, K. S., Grini, A., Hamilton, D. S., Johnson, J. S., Karydis, V. A., Kirkevåg, A., Kodros, J. K., Lohmann, U., Luo, G., Makkonen, R., Matsui, H., Neubauer, D., Pierce, J. R., Schmale, J., Stier, P., Tsigaridis, K., Noije, T. van, Wang, H., Watson-Parris, D., Westervelt, D. M., Yang, Y., Yoshioka, M., Daskalakis, N., Decesari, S., Gysel Beer, M., Kalivitis, N., Liu, X., Mahowald, N. M., Myriokefalitakis, S., Schrödner, R., Sfakianaki, M., Tsimpidi, A. P., Wu, M. and Yu, F.: Evaluation of global simulations of aerosol particle number and

cloud condensation nuclei, and implications for cloud droplet formation, Atmospheric Chemistry and Physics Discussions, 1–40, doi:https://doi.org/10.5194/acp-2018-1340, 2019.

Fuller, K. A., Malm, W. C. and Kreidenweis, S. M.: Effects of mixing on extinction by carbonaceous particles, Journal of Geophysical Research: Atmospheres, 104(D13), 15941–15954, doi:10.1029/1998JD100069, 1999.

Hammer, E., Bukowiecki, N., Gysel, M., Jurányi, Z., Hoyle, C. R., Vogt, R., Baltensperger, U. and Weingartner, E.: Investigation of the effective peak supersaturation for liquid-phase clouds at the high-alpine site Jungfraujoch, Switzerland (3580 m a.s.l.), Atmospheric Chemistry and Physics, 14(2), 1123–1139, doi:10.5194/acp-14-1123-2014, 2014.

Hoyle, C. R., Webster, C. S., Rieder, H. E., Nenes, A., Hammer, E., Herrmann, E., Gysel, M., Bukowiecki, N., Weingartner, E., Steinbacher, M. and Baltensperger, U.: Chemical and physical influences on aerosol activation in liquid clouds: a study based on observations from the Jungfraujoch, Switzerland, Atmospheric Chemistry and Physics, 16(6), 4043–4061, doi:10.5194/acp-16-4043-2016, 2016.

Jurányi, Z., Gysel, M., Weingartner, E., Bukowiecki, N., Kammermann, L. and Baltensperger, U.: A 17 month climatology of the cloud condensation nuclei number concentration at the high alpine site Jungfraujoch, Journal of Geophysical Research, 116(D10), doi:10.1029/2010JD015199, 2011.

Kammermann, L., Gysel, M., Weingartner, E. and Baltensperger, U.: 13-month climatology of the aerosol hygroscopicity at the free tropospheric site Jungfraujoch (3580 m a.s.l.), Atmospheric Chemistry and Physics, 10(22), 10717–10732, doi:https://doi.org/10.5194/acp-10-10717-2010, 2010.

Luo, J., Zhang, Y., Wang, F. and Zhang, Q.: Effects of brown coatings on the absorption enhancement of black carbon: a numerical investigation, Atmospheric Chemistry and

Physics, 18(23), 16897–16914, doi:https://doi.org/10.5194/acp-18-16897-2018, 2018.

Schmale, J., Henning, S., Decesari, S., Henzing, B., Keskinen, H., Sellegri, K., Ovad-nevaite, J., Pöhlker, M. L., Brito, J., Bougiatioti, A., Kristensson, A., Kalivitis, N., Stavroulas, I., Carbone, S., Jefferson, A., Park, M., Schlag, P., Iwamoto, Y., Aalto, P., Äijälä, M., Bukowiecki, N., Ehn, M., Frank, G., Fröhlich, R., Frumau, A., Herrmann, E., Herrmann, H., Holzinger, R., Kos, G., Kulmala, M., Mihalopoulos, N., Nenes, A., O'Dowd, C., Petäjä, T., Picard, D., Pöhlker, C., Pöschl, U., Poulain, L., Prévôt, A. S. H., Swietlicki, E., Andreae, M. O., Artaxo, P., Wiedensohler, A., Ogren, J., Matsuki, A., Yum, S. S., Stratmann, F., Baltensperger, U. and Gysel, M.: Long-term cloud condensation nuclei number concentration, particle number size distribution and chemical composition measurements at regionally representative observatories, Atmospheric Chemistry and Physics, 18(4), 2853–2881, doi:10.5194/acp-18-2853-2018, 2018.

Taylor, J. W., Allan, J. D., Liu, D., Flynn, M., Weber, R., Zhang, X., Lefer, B. L., Grossberg, N., Flynn, J. and Coe, H.: Assessment of the sensitivity of core/shell parameters derived using the single-particle soot photometer to density and refractive index, Atmospheric Measurement Techniques, 8(4), 1701–1718, doi:10.5194/amt-8-1701-2015, 2015.

Verheggen, B., Cozic, J., Weingartner, E., Bower, K., Mertes, S., Connolly, P., Gallagher, M., Flynn, M., Choularton, T. and Baltensperger, U.: Aerosol partitioning between the interstitial and the condensed phase in mixed-phase clouds, Journal of Geophysical Research-Atmospheres, 112, doi:10.1029/2007jd008714, 2007.